# The unique synaptic circuitry of specialized olfactory glomeruli in *Drosophila melanogaster*

**Lydia Gruber[1], Rafael Cantera[2], Markus William Pleijzier[3], Martin Niebergall[1], Michael Steinert[4], Thomas Pertsch[4], Bill S Hansson[1†], Jürgen Rybak[1*†]**

[1]Max Planck Institute for Chemical Ecology, Department of Evolutionary Neuroethology, Jena, Germany; [2]Instituto de Investigaciones Biológicas Clemente Estable, Departamento de Biología del, Montevideo, Uruguay; [3]Neurobiology Division, MRC Laboratory of Molecular Biology, Cambridge, United Kingdom; [4]Institute of Applied Physics, Abbe Center of Photonics, Friedrich Schiller University Jena, Jena, Germany

**\*For correspondence:**
jrybak@ice.mpg.de

[†]These authors contributed equally to this work

**Competing interest:** The authors declare that no competing interests exist.

## eLife Assessment

This study seeks to determine how synaptic relationships between principal cell types in the olfactory system vary with glomerulus selectivity and is therefore **valuable** to the field. The methodology is **solid**, and with the caveat that here was a technical need to group all local interneurons, centrifugal neurons and multiglomerular projection neurons into one category ("multiglomerular neurons"), this work reveals some very interesting potential differences in circuit architecture associated with glomerular tuning breadth.

**Abstract** In the *Drosophila* olfactory system, most odorants are encoded in the antennal lobe in a combinatory way, activating several glomerular circuits. However, odorants of particular ecological role for the fly are encoded through activation of a single specialized olfactory pathway. Comparative analyses of densely reconstructed connectomes of one broadly tuned glomerulus (DL5) and one narrowly tuned glomerulus (DA2) gained detailed insight into the variations of synaptic circuitries of glomeruli with different computational tasks. Our approach combined laser branding of glomeruli of interest with volume-based focused ion beam-scanning electron microscopy to enable precise targeting and analysis of the two glomeruli. We discovered differences in their neuronal innervation, synaptic composition, and specific circuitry of their major cell types: olfactory sensory neurons (OSNs), uniglomerular projection neurons, and multiglomerular neurons. By comparing our data with a previously mapped narrowly tuned glomerulus (VA1v), we identified putative generic features of narrowly tuned glomerular circuits, including higher density of neuronal fibers and synapses, lower degree of OSN lateralization, stronger axo-axonic connections between OSNs, dendro-dendritic connections between many uPNs, and lower degree of presynaptic input on OSN axons. In addition, this work revealed that the dendrites of the single uPN in DL5 contain a substantial amount of autapses interconnecting distant regions of the dendritic tree. The comparative analysis of glomeruli allows us to formulate synaptic motifs implemented in olfactory circuits with different computational demands.

## Introduction

Olfaction is an anatomically shallow sensory system. In mammals and invertebrates, just one synapse separates the sensory periphery from the central brain (*Su et al., 2009*; *Liang and Luo, 2010*; *Shepherd, 2011*; *Owald and Waddell, 2015*; *Dolan et al., 2018*). In the olfactory system of *Drosophila*

*melanogaster*, the first relay station of synaptic transmission is the antennal lobe (AL), which has a circuit architecture homologous to that of the vertebrate olfactory bulb (*Boeckh et al., 1990*; *Sachse and Manzini, 2021*; *Shepherd et al., 2021*). The fly AL consists of approximately 58 spherical compartments, called glomeruli, which can be distinguished by size, shape, and location (*Laissue et al., 1999*; *Gao et al., 2000*; *Vosshall et al., 2000*; *Grabe et al., 2015*; *Bates et al., 2020*). Each glomerulus receives stereotypic input from axon terminals of olfactory sensory neurons (OSNs), which have their cell bodies and dendrites located in the antennae or maxillary palps (*De Bruyne et al., 1999*; *Shanbhag et al., 1999*; *De Bruyne et al., 2001*; *Hallem et al., 2004*; *Benton et al., 2006*). All the OSNs innervating a given glomerulus express a typical repertoire of ligand-gated chemoreceptors (*Couto et al., 2005*; *Fishilevich and Vosshall, 2005*; *Benton et al., 2006*), which represent a wide range of specifications, binding either a single, few, or many distinct chemicals (*Hallem et al., 2004*; *Hallem and Carlson, 2006*; *Knaden et al., 2012*; *Münch and Galizia, 2016*; *Seki et al., 2017*; *Wicher and Miazzi, 2021*).

Most OSNs project bilaterally to the corresponding glomeruli in the left and right AL (*Gaudry et al., 2013*; *Tobin et al., 2017*). In the AL, OSNs convey odor signals to excitatory uniglomerular projection neurons (uPNs), which branch only within a single glomerulus, or to inhibitory multiglomerular PNs (mPNs) and inhibitory or excitatory local interneurons (LNs) (*Ng et al., 2002*; *Cuntz et al., 2007*; *Kazama and Wilson, 2008*; *Kreher et al., 2008*; *Kazama and Wilson, 2009*; *Masse et al., 2009*; *Tanaka et al., 2012*; *Ai and Hagio, 2013*; *Wilson, 2013*; *Bates et al., 2020*). LNs innervate each several glomeruli and are the key modulatory neurons in the AL (*Chou et al., 2010*; *Seki et al., 2010*). The highly converging OSN-to-PN signal transmission (*Chen and Shepherd, 2005*; *Masse et al., 2009*; *Jeanne and Wilson, 2015*) is lateralized, activating ipsilateral uPNs more strongly than contralateral ones (*Agarwal and Isacoff, 2011*; *Gaudry et al., 2013*; *Tobin et al., 2017*). From the AL, uPNs and mPNs relay processed signal information to higher brain centers (*Norgate et al., 2006*; *Fiala, 2007*; *Jefferis et al., 2007*; *Keene and Waddell, 2007*; *Galizia, 2014*; *Guven-Ozkan and Davis, 2014*; *Strutz et al., 2014*; *Bates et al., 2020*).

The stereotypic activity pattern elicited by distinct odorants encodes the odor space, represented in a so-called odotopic map of the AL according to the glomerular activation by distinct chemical classes (*Couto et al., 2005*; *Laissue and Vosshall, 2008*; *Knaden and Hansson, 2014*; *Grabe et al., 2015*; *Grabe and Sachse, 2018*). Some odorants induce a fixed innate behavior (aversion or attraction), activating characteristically specific glomeruli (*Semmelhack and Wang, 2009*; *Knaden et al., 2012*; *Knaden and Hansson, 2014*; *Gao et al., 2015*; *Grabe and Sachse, 2018*). The encoding of hedonic valence already at the level of the AL is important for a fast odor coding. Most odorants are encoded in a combinatorial manner in the fly AL by activating multiple OSN types expressing broadly tuned receptors and their glomerular circuits, including broadly tuned uPNs (*De Bruyne et al., 2001*; *Silbering and Galizia, 2007*; *Silbering et al., 2008*; *Masse et al., 2009*; *Galizia, 2014*; *Szyszka and Galizia, 2015*; *Sachse and Hansson, 2016*; *Seki et al., 2017*). Certain chemoreceptors and their downstream glomerular circuits, however, have evolved a very high specificity and sensitivity to single or very few chemicals (*Andersson et al., 2015*; *Haverkamp et al., 2018*; *Keesey and Hansson, 2021*). These narrowly tuned glomerular circuits often belong to dedicated olfactory pathways, called 'labeled lines', which process information regarding single odorants of particular importance for reproduction and survival (*Kurtovic et al., 2007*; *Datta et al., 2008*; *Stensmyr et al., 2012*; *Dweck et al., 2015*; *Gao et al., 2015*). An extreme example is the DA2 glomerulus, which responds exclusively to geosmin, an ecologically relevant chemical that alerts flies to the presence of harmful microbes, causing the fly to avoid laying eggs at these locations (*Stensmyr et al., 2012*). This dedicated olfactory pathway and its receptor sequence are conserved throughout evolution (*Keesey et al., 2019*; *Keesey and Hansson, 2021*). Another example is glomerulus VA1v, which responds to methyl laurate, a pheromone that induces a strongly attractive response in female flies leading to aggregation behavior (*Dweck et al., 2015*). DL5, on the other hand, is an example of a broadly tuned glomerulus, innervated by OSNs activated by several odorants, like E2-hexenal and benzaldehyde (*Knaden et al., 2012*; *Münch and Galizia, 2016*; *Seki et al., 2017*; *Mohamed et al., 2019b*). This functional diversity suggests differences in neuronal composition and synaptic connectivity between broadly and narrowly tuned glomeruli.

A survey of neuronal composition across glomeruli revealed great variation in the numbers of the different types of neurons innervating narrowly and broadly tuned glomeruli (*Grabe et al., 2016*).

In general, narrowly tuned glomeruli are innervated by more uPNs and fewer LNs compared with more broadly tuned glomeruli (*Chou et al., 2010*; *Grabe et al., 2016*). In addition, narrowly tuned OSNs receive less global LN inhibition than broadly tuned ones (*Hong and Wilson, 2015*; *Grabe et al., 2020*; *Schlegel et al., 2021*). For example, in female flies, the narrowly tuned glomerulus DA2 contains dendrites of six to eight uPNs, whereas the broadly tuned glomerulus DL5 houses only one or two uPNs and has a higher number of innervating LNs. Interestingly, both glomeruli are innervated by the same number of OSNs (*Grabe et al., 2016*).

Little is known, however, about the microarchitecture of the synaptic circuitry in distinct glomeruli and, in particular, about ultrastructural differences between narrowly vs. broadly tuned glomerular circuits. Electron microscopy (EM) allows volume imaging with dense reconstruction of fine neurite branches and synapses in brain tissue at nanometer resolution, necessary to map synapses (*Briggman and Denk, 2006*; *Cardona et al., 2009*; *Helmstaedter, 2013*; *Rybak, 2013*; *Meinertzhagen, 2018*). The first ultrastructural insights into the synaptic connectivity of *Drosophila* olfactory glomeruli were obtained by studies based on serial section transmission EM (ssTEM) (*Rybak, 2016a*; *Tobin et al., 2017*). *Rybak et al., 2016b* showed that all three basic classes of AL neurons make synapses with each other, while *Tobin et al., 2017* revealed that the differences in number of innervating uPNs between the left and right DM6 glomeruli are compensated by differences in synaptic strength. With focused ion beam-scanning electron microscopy (FIB-SEM; *Knott et al., 2008*), a complete reconstruction of all neurons in the narrowly tuned, pheromone processing glomerulus VA1v was obtained (*Horne et al., 2018*). Recent technological innovations in ssTEM, FIB-SEM, and automated neuron reconstruction have made connectome datasets of the adult *Drosophila* central nervous system available (*Saalfeld et al., 2009*; *Zheng et al., 2018*; *Li et al., 2020b*; *Scheffer et al., 2020*) and provided complete circuit descriptions of several brain centers (*Felsenberg et al., 2018*; *Dolan et al., 2019*; *Auer et al., 2020*; *Bates et al., 2020*; *Coates et al., 2020*; *Huoviala et al., 2020*; *Li et al., 2020b*; *Li et al., 2020a*; *Marin et al., 2020*; *Otto et al., 2020*; *Hulse et al., 2021*; *Schlegel et al., 2021*).

To understand how highly specialized glomerular circuits in dedicated olfactory pathways differ in signal integration from broadly tuned circuits, we compared the microarchitecture and synaptic connectivity of a narrowly tuned glomerulus (DA2) and a broadly tuned glomerulus (DL5).

By using a correlative workflow that combines transgenic markers with FIB-SEM to identify these glomeruli, we reconstructed OSNs, uPNs, and multiglomerular neurons (MGNs), comprising both LNs and mPNs of the AL combined. We mapped all associated synapses and compared the circuit organization of both glomeruli.

## Results

### Volume-based EM of two different olfactory glomeruli

To compare the synaptic circuitries of two olfactory glomeruli known to belong to either narrowly or broadly tuned glomerular types in *D. melanogaster*, we mapped all synapses of glomeruli DA2 (right AL) and DL5 (left AL) in a single female fly (*Figure 1A, B*) with the aid of FIB-SEM. A partial reconstruction of a second DA2 in another fly was used to measure neuronal volume (see Methods). The reconstructions were based on high resolution (4 × 4 × 20 nm) datasets (*Figure 1*; *Figure 1—video 1*), thus allowing reconstruction of the finest neuronal branches (~20 nm diameter; *Figure 1C, D*) as well as mapping chemical synapses (example in *Figure 1E*) in the two volumes of interest (VOIs). To restrict the imaging volume to the target VOIs, we employed a correlative approach for the first time for a *Drosophila* EM volume reconstruction. Glomeruli DA2 and DL5 were identified by their size, shape, and location in brains of transgenic flies (*Orco-GAL4; UAS-GCaMP6s*) using the glomerular map of *Grabe et al., 2015*. The flies expressed the green fluorescent protein GCaMP6 coupled with calmodulin and M13 (a peptide sequence from myosin light-chain kinase; *Figure 1A, B*). Subsequently, the identified glomeruli were marked by laser branding using a two-photon laser (*Bishop et al., 2011*). These fiducial marks were apparent under both light microscopy (*Figure 1A, B*) and EM (*Figure 1C, D*) and facilitated the delimitation of the VOIs during FIB-SEM scanning. We produced two complete FIB-SEM datasets: one for glomerulus DA2 and one for DL5 (pure imaging time for both glomeruli: ~60 hr) and a partial dataset for DA2 in a second fly, which was used for volumetric analysis (see data availability).

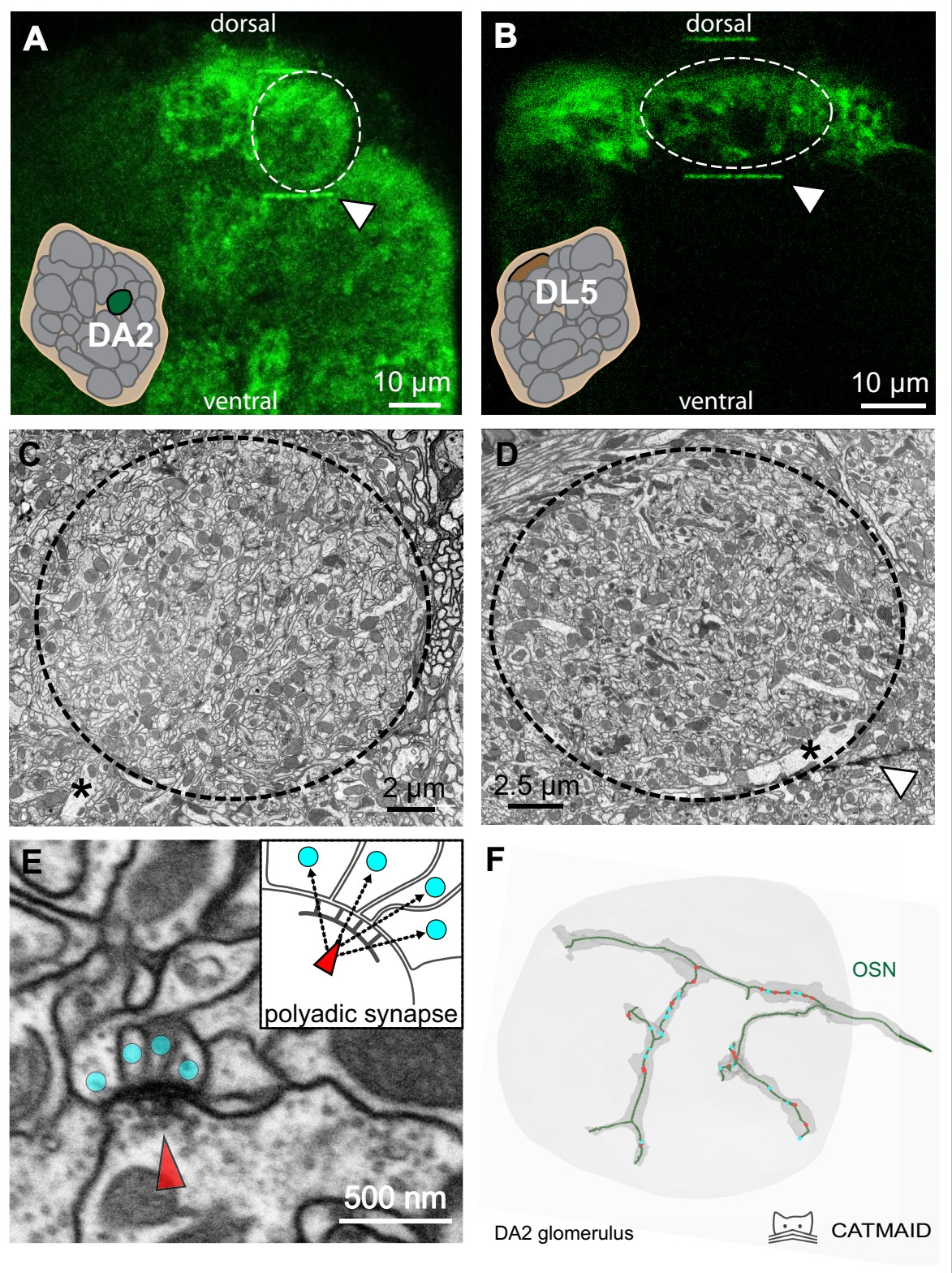

**Figure 1.** A correlative approach to analyze the ultrastructure of identified olfactory glomeruli. (**A, B**) Two-photon laser scans of the antennal lobe in *Orco-Gal4; UAS-GCaMP6s* flies where Orco-positive olfactory sensory neurons (OSNs) in the glomerular neuropil were labeled by GCaMP (green fluorescence). Glomeruli DA2 (**A**) and DL5 (**B**) are encircled. Schematics show their relative position in the antennal lobe. Once the glomerulus of interest was identified, its boundaries were delineated using fiducial marks (white triangles) via laser branding, which enabled their identification at the ultrastructural level. Representative images of the same glomeruli (DA2 in **C** and DL5 in **D**) obtained with focused ion beam-scanning electron

*Figure 1 continued on next page*

*Figure 1 continued*

microscopy (FIB-SEM), showing their ultrastructure. Asterisks indicate the main neurite of uniglomerular projection neurons entering the glomerulus. White triangle shows a two-photon laser mark (see also **A** and **B**). (**E**) FIB-SEM image of a polyadic synapse: the presynaptic site (red arrowhead) is composed of a T-bar shaped presynaptic density, surrounded by small vesicles, and is opposed to several postsynaptic profiles (cyan dots). Scheme of a tetrad synapse: a presynaptic site with its T-bar (red arrowhead) forms four output connections (arrows) with four postsynaptic input sites (cyan dots). (**F**) A skeleton-based reconstruction of an OSN axon terminal (green line) with presynaptic (red dots) and postsynaptic sites (cyan dots). The dark gray shading surrounding the OSN trace represents the volume-based reconstruction of the same neuron. Tracing and reconstruction were performed within the FIB-SEM dataset (light gray area).

The online version of this article includes the following video for figure 1:

**Figure 1—video 1.** Focused ion beam-scanning electron microscopy (FIB-SEM) dataset of a DA2 glomerulus featuring a uniglomerular projection neuron (uPN) reconstruction (see extra file).

https://elifesciences.org/articles/88824/figures#fig1video1

## Skeleton-based neuron reconstruction and synapse identification

We reconstructed all neurons within the two VOIs (example neuron: *Figure 1F*) and mapped all their synaptic connections using an iterative skeleton-based reconstruction approach, similar to previously reported procedures (*Berck et al., 2016*; *Schneider-Mizell et al., 2016*; *Zheng et al., 2018*) with the aid of the web-based neuron reconstruction software CATMAID (https://catmaid.readthedocs.io/en/stable/); RRID:SCR_006278; (*Cardona et al., 2009*; *Schneider-Mizell et al., 2016*; *Figure 1—video 1*). Synapses were identified by their presynaptic transmitter release site, which in *Drosophila* is composed of a presynaptic density called a T-bar, surrounded by synaptic vesicles and apposed post-synaptic elements (*Figure 1E*), as previously described (*Trujillo-Cenóz, 1969*; *Fröhlich, 1985*; *Rybak et al., 2016b*; *Huang et al., 2018*; *Li et al., 2020b*). All synapses observed in our FIB-SEM datasets were polyadic, that is, each presynaptic site connected to multiple postsynaptic sites (see example in *Figure 1E*), a feature of insect brain synapses (*Meinertzhagen and O'Neil, 1991*; *Malun et al., 1993*; *Prokop and Meinertzhagen, 2006*; *Hartenstein, 2016*; *Rybak et al., 2016b*). Some synapses had up to 16 postsynaptic sites, that is, one T-bar and 16 single synaptic profiles (i.e., sixteen 1:1 single output–input connections). Short neuronal fragments (<10 μm), which could not be connected to any neuronal fiber were designated as 'orphans'. These fragments represented 4% of the total length of all traced neuronal fibers in DA2 and 6% in DL5 and contained about ~12% of all synaptic contacts in both glomeruli.

## Glomerular neurons: classification, description, and inventory

Previous descriptions of the ultrastructural characteristics of the AL in *Drosophila* helped to classify AL neurons into three main classes (*Figure 2A*): OSNs, uPNs, and MGNs (cells that interconnect multiple glomeruli). MGNs are further subdivided into mPNs and LNs (*Berck et al., 2016*; *Rybak et al., 2016b*; *Gruber et al., 2018*; *Horne et al., 2018*; *Zheng et al., 2018*; *Schlegel et al., 2021*). Most of the neuronal profiles within the MGN neuron class probably belong to inhibitory local neurons, as this cell type is the most numerous and broadly arborizing of the multiglomerular cell types in the AL (*Chou et al., 2010*; *Lin et al., 2012*). In addition, we observed a few neuronal fibers with an electron-dense and vesicle-rich cytosol, which we interpreted to be either peptidergic neurons (*Nässel and Homberg, 2006*; *Eckstein et al., 2024*) or the contralaterally projecting, serotonin-immunoreactive deutocerebral (CSD) neuron (*Dacks et al., 2006*; *Goyal and Chaudhury, 2013*; *Zheng et al., 2018*; *Coates et al., 2020*; *Eckstein et al., 2024*). Except for these neuronal fibers containing abundant electron-dense vesicles, all other neuronal fibers were assigned to either OSNs, uPNs, or MGNs based on their morphology (*Figure 2A, B*; see Methods).

   OSNs formed large, elongated synaptic boutons (*Figure 2A*), had the largest volume/length ratio of all three neuron classes (*Figure 2—figure supplement 1*) and displayed the lowest degree of branching intensity of all neurons in both glomeruli (*Figure 2B*). In agreement with what had been observed in other glomeruli (*Rybak et al., 2016b*), the majority of output synapses formed by OSN terminals were triads (1:3) and tetrads (1:4). However, the T-bars of OSN synapses showed consider-able size variation, with some of them large enough to accommodate up to 16 postsynaptic contacts (*Figure 2—figure supplement 1*). The frequency of large T-bars was much higher in OSNs than in other neuron classes with an average polyadicity (average number of postsynaptic sites at each T-bar) of 6 (1:6; *Table 1*, row 14). As OSNs had the greatest T-bar and output density along their axons

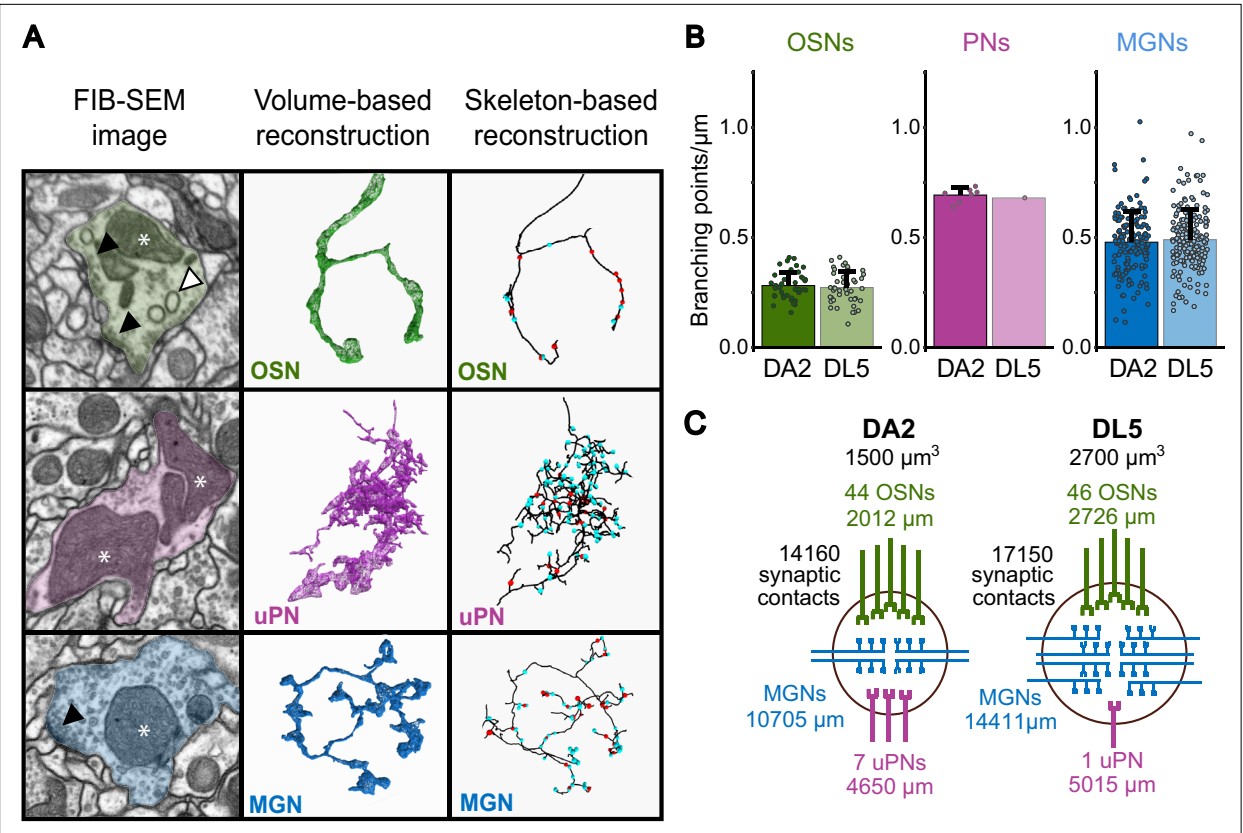

**Figure 2.** Neuron classification and neuronal composition of the DA2 and DL5 glomeruli. (**A**) Representative examples of each neuron class in glomeruli DA2 and DL5. Shown are focused ion beam-scanning electron microscopy (FIB-SEM) images (left column), volumetric neuronal reconstructions (middle column), and skeleton-based neuron traces (right column) for olfactory sensory neurons (OSNs, green), uniglomerular projection neurons (uPNs, magenta), and multiglomerular neurons (MGNs, blue). Key ultrastructural features, including T-bars (black triangle), mitochondria (asterisks), and spinules (white triangle) are indicated on the FIB-SEM images. Volumetric reconstructions (middle column) depict the general morphology of each neuron class. On the skeleton traces, pre- and postsynaptic sites are marked with red and cyan dots, respectively (right column). (**B**) Average branching intensity (branching points per μm of neuronal fiber length) for each neuron class OSNs, uPNs, and MGNs in DA2 and DL5. Data represent mean + standard deviation (error bars). Data points represent single values. Means were compared using the Wilcoxon two-sample test. No significant differences in branching points/μm were observed for OSNs or MGNs between glomeruli (significance was not tested for uPNs due to the presence of a single uPN in DL5). (**C**) Schematic summary, for each glomerulus, showing its volume (in μm³), the number of neurons per class (excluding MGNs), the total fiber length of all neurons for each neuron class and the total number of individual synaptic contacts.

The online version of this article includes the following source data and figure supplement(s) for figure 2:

**Source data 1.** Branching point analysis of olfactory sensory neurons (OSNs), uniglomerular projection neurons (uPNs), and multiglomerular neurons (MGNs) in DA2 and DL5 (X_DA2_rAl and X_DL5_lAl).

**Figure supplement 1.** Neuronal volume and polyadicity.

(*Table 1*, rows 10 and 11) they also displayed the largest synaptic ratios (both for the T-bars/input sites and output sites/input sites) of all neuron classes (*Table 1*, rows 12 and 13), which was in line with previous observations (*Rybak et al., 2016b*).

The uPNs exhibited the highest degree of branching intensity of the three neuron classes in both glomeruli (*Figure 2A, B*). They showed numerous very fine apical branches that frequently connected multiple times via spines to the same presynaptic site, leading to an entangled 3D shape typical for this neuron class (*Figure 2A*; *Rybak et al., 2016b*; *Tobin et al., 2017*; *Schlegel et al., 2021*). uPNs had the smallest volume/length ratio of all neuron classes (for the DA2: *Figure 2—figure supplement 1A*). In addition to having many fine branches, uPN dendrites also had enlarged regions with almost no cytosol that were packed with large mitochondrial profiles extending over considerable distances. These enlarged profiles showed a larger degree of mitochondria fission (dividing and segregating mitochondrion organelles; personal observation) than the other neuron classes with rather round and compact mitochondria (*Figure 2A*; FIB-SEM image; see data availability). Seven uPNs were found in

**Table 1.** Glomerular innervation and synaptic composition.

Quantitative comparison of neuronal and synaptic parameters between glomeruli DA2 and DL5 for each neuron class – olfactory sensory neurons (OSNs, green), uniglomerular projection neurons (uPNs, magenta), and multiglomerular neurons (MGNs, blue) – as well as the combined totals. **Row 1**: Total length of all neurons per neuron class and the overall total length for all neurons for each glomerulus. **Rows 2–4**: Synapse counts: number of input sites (inputs), output sites (outputs), and T-bars. **Row 5**: Innervation density: calculated as total neuron length (µm; row 1)/glomerular volume (µm$^3$); glomerular volumes: DA2 = 1500 µm$^3$ and DL5 = 2600 µm$^3$ (see *Figure 1C*). **Rows 6–8**: Synaptic density per unit glomerular volume (µm$^3$): total number of all input sites (inputs), output sites (outputs), and T-bars for each neuron class and all neurons divided by glomerular volume. **Rows 9–11**: Average synaptic density along neuronal fibers (see also *Figure 3—figure supplement 1*): number of inputs, outputs, or T-bars per µm of neuron length. **Rows 12 and 13**: Average synaptic ratios: T-bars-to-inputs or outputs-to-inputs. **Row 14**: Polyadicity: the average number of postsynaptic sites at each T-bar in DA2 and DL5. The ratios in rows 12–14 were calculated based on synaptic counts normalized to neuron length (rows 9–11). The color shading highlights values with a relative difference greater than 20% between DA2 and DL5 (see relative differences, *Supplementary file 1*). Highlighted values greater in DA2 than DL5 are underlined.

| Row | Values | Unit | OSNs | | uPNs | | MGNs | | All neurons | |
|---|---|---|---|---|---|---|---|---|---|---|
| | | | DA2 | DL5 | DA2 | DL5 | DA2 | DL5 | DA2 | DL5 |
| 1 | Total neuronal length | µm | 2012 | 2727 | 4652 | 5015 | 10,705 | 14,411 | 17,370 | 22,153 |
| 2 | | inputs | 868 | 1083 | 3887 | 3955 | 7229 | 9018 | 11,984 | 14,056 |
| 3 | | outputs | 6671 | 6828 | 1624 | 3108 | 5659 | 6749 | 13,954 | 16,685 |
| 4 | Total synaptic counts | T-bars | 1063 | 1213 | 322 | 602 | 1263 | 1572 | 2648 | 3387 |
| 5 | Total innervation density (sum of length of all neuronal fibers/glomerular volume) | µm/µm$^3$ | 1.26 | 1.05 | 2.91 | 1.93 | 6.69 | 5.54 | 10.86 | 8.52 |
| 6 | | inputs/µm$^3$ | 0.54 | 0.42 | 2.43 | 1.52 | 4.52 | 3.47 | 7.49 | 5.41 |
| 7 | | outputs/µm$^3$ | 4.17 | 2.63 | 1.02 | 1.20 | 3.54 | 2.60 | 8.72 | 6.42 |
| 8 | Total glomerular synaptic density (total synaptic counts/ glomerular volume) | T-bars/µm$^3$ | 0.66 | 0.47 | 0.20 | 0.23 | 0.79 | 0.60 | 1.66 | 1.30 |
| 9 | | inputs/µm | 0.42 | 0.39 | 0.83 | 0.79 | 0.62 | 0.59 | 0.59 | 0.56 |
| 10 | | outputs/µm | 3.37 | 2.62 | 0.33 | 0.62 | 0.52 | 0.51 | 1.06 | 0.87 |
| 11 | Neuronal synaptic density (synaptic counts/neuronal length) | T-bars/µm | 0.53 | 0.46 | 0.07 | 0.12 | 0.12 | 0.12 | 0.19 | 0.18 |
| 12 | | T-bars/inputs | 1.31 | 1.27 | 0.08 | 0.15 | 0.23 | 0.24 | 0.43 | 0.42 |
| 13 | Synaptic ratio | outputs/inputs | 8.29 | 7.29 | 0.40 | 0.79 | 1.04 | 1.11 | 2.40 | 2.17 |
| 14 | Polyadicity | outputs/T-bars | 6.35 | 5.70 | 4.95 | 5.16 | 3.23 | 2.64 | 3.85 | 3.17 |

The online version of this article includes the following source data for table 1:

**Source data 1.** Measurements and analysis of neuronal features (olfactory sensory neurons [OSNs], uniglomerular projection neurons [uPNs], and multiglomerular neurons [MGNs]) in DA2 and DL5 (X_DA2_rAl and X_DL5_lAl), summarized in *Table 1*.

DA2, confirming light microscopy findings (*Grabe et al., 2016*). Two of them (PN#1, PN#2; see data availability) branched broadly and innervated the full glomerulus and received more synaptic input than the other 5 uPNs (PN#3–#7; see *Figure 5—source data 3*), which branched exclusively in sub-regions of the glomerulus, with partial overlap. In addition to abundant clear small vesicles (~20 nm in diameter) (*Yasuyama et al., 2003*; *Strutz et al., 2014*; *Bates et al., 2020*), uPN dendrites also displayed small electron-dense vesicles, as previously reported for PN axon terminals in the mushroom body calyx (*Butcher et al., 2012*; *Yang et al., 2022*). These electron-dense vesicles are packed with different types of neuropeptides that act as neuromodulators or co-transmitters (*Gondré-Lewis et al., 2012*; *Croset et al., 2018*; *Eckstein et al., 2024*). In both glomeruli, uPNs had the highest neuronal synaptic input density and the lowest T-bar and output density of the three neuron classes (*Table 1*, rows 9–11; DA2 and DL5 differences: see next section). The synaptic ratios (T-bars/input sites and output sites/input sites) were much lower for uPNs than for the other neuron classes (*Table 1*, rows 12 and 13). The majority of uPN dendritic output synapses (feedback synapses) were tetrads in both glomeruli, with an average polyadicity of around 5 lower than in OSNs; (*Figure 2—figure supplement 1*; *Table 1*, row 14).

The majority of the neuronal fibers in both glomeruli belonged to MGNs (*Figure 2A*). MGNs exhibited variable morphology and ultrastructure, as expected, but also shared some ultrastructural features. Their synaptic boutons were formed by thin fibers, thus the volume/length ratio of MGNs was lower than that of OSNs but greater than that of uPNs (*Figure 2—figure supplement 1A*). A similar relationship was found for the number of output sites and the T-bar density along MGN fibers, which were smaller than in OSNs but larger than in uPNs (*Table 1*, rows 10 and 11). In contrast, branching intensity in MGNs was larger than in OSNs but smaller than in uPNs (*Figure 2B*). The synaptic ratio of output-to-input sites was around one (*Table 1*, rows 12 and 13). MGNs had the lowest polyadicity (~3) of the three neuron classes (*Table 1*, row 14) and their synapses were mainly triads. Interestingly, besides the abundant clear small vesicles (~20 nm in diameter), some MGNs had small electron-dense vesicles, most likely housing neuropeptides (*Carlsson et al., 2010*; *Croset et al., 2018*; *Nässel, 2018*).

## DA2 is more densely innervated and has a higher synapse density than DL5

In our FIB-SEM datasets, the volume of glomerulus DA2 was 45% smaller than that of glomerulus DL5 (1500 vs. 2640 $\mu m^3$), which is in agreement with measurements based on light microscopy DA2 = 1600 $\mu m^3$, DL5 = 2900 $\mu m^3$ (*Grabe et al., 2016*). We also confirmed that a similar number of OSNs (44–46 OSNs) innervated both glomeruli (*Figure 2C*), and that each glomerulus received input from OSNs originating in both the ipsilateral and contralateral antennae (*Vosshall et al., 2000*; *Grabe et al., 2016*). Consistent with previous findings (*Grabe et al., 2016*), the DA2 glomerulus was innervated by seven uPNs, whereas DL5 had a single uPN (*Figure 2C*). MGN cell numbers could not be determined in our study due to their multiglomerular morphology, which also prevented us from tracing MGN fibers to their soma due to our partial volume acquisition (see Methods).

To investigate differences between DA2 and DL5, we turned our attention to their glomerular innervation and synaptic composition. We measured the total length (sum in $\mu m$) of all neuronal fibers of each neuron class within the DA2 and DL5 (*Figure 2C*; *Table 1*, row 1). In addition, we counted all T-bars and their output sites (1:1 synaptic contacts) as well as all postsynaptic sites (input sites) for all neuron fibers together and for each neuron class individually (*Table 1*, rows 2–4). We counted in total ~14,000 synaptic contacts and 2648 T-bars in DA2 and ~17,000 contacts and 3387 T-bars in DL5 (*Figure 2C*, *Table 1*, row 4). Most of these synapses were triads and tetrads (*Figure 2—figure supplement 1B–D*). In order to compare DA2 and DL5, we normalized neuronal length and synaptic numbers to glomerular volume. We then analyzed (1) the innervation density, that is, the length of neuronal fibers per glomerular volume ($\mu m/\mu m^3$) and (2) the glomerular synaptic density (T-bar #, output site or input site #/$\mu m^3$). Data are reported in total for all neuronal fibers of each neuron class (*Table 1*, rows 5–8) and as an average for neuronal fibers of the respective neuron class (*Figure 3*). In addition, we compared (3) the average polyadicity for each neuron class (*Figure 3*) and (4) the average neuronal synaptic density (T-bar, output, and input site density along each neuronal fiber) (#/$\mu m$) (*Figure 3—figure supplement 1*).

We observed that the average neuron innervation density of OSNs was 20% higher in DA2 than in DL5 (*Figure 3A*, *Supplementary file 1*). Also, the glomerular synaptic density of input sites, output sites, and T-bars along OSNs was significantly higher in DA2 than in DL5 (*Figure 3A*). OSNs in DA2 formed therefore more input sites, and much more T-bars and output sites per glomerular volume than in DL5 (*Table 1*, rows 7 and 8; relative differences: *Supplementary file 1*). In contrast, the density of input sites distributed along the length of OSN fibers was similar in DA2 and DL5, whereas T-bar and output site density along the OSN axons was significantly higher in DA2 (*Figure 3—figure supplement 1B*).

We then asked if the DA2 glomerulus, due to its higher number of uPNs, also had a higher uPN innervation density and synaptic density of its postsynaptic sites and/or presynaptic sites compared to the DL5 glomerulus, which contains a single uPN. In the DA2, the fibers of the seven uPNs had almost the same total length as the fibers of the single uPN in the more voluminous DL5 (4652 $\mu m$ in DA2 vs. 5015 $\mu m$ in DL5; *Table 1*, row 1). The DA2 uPNs had, in addition, a total number of input sites comparable to that of the single uPN in DL5 (3887 vs. 3955; *Table 1*, row 2). Consequently, the total innervation density of the seven DA2-uPNs was higher than that of the single uPN in DL5 (*Table 1*, row 5), even though the average innervation density per DA2-uPN was lower compared with that of

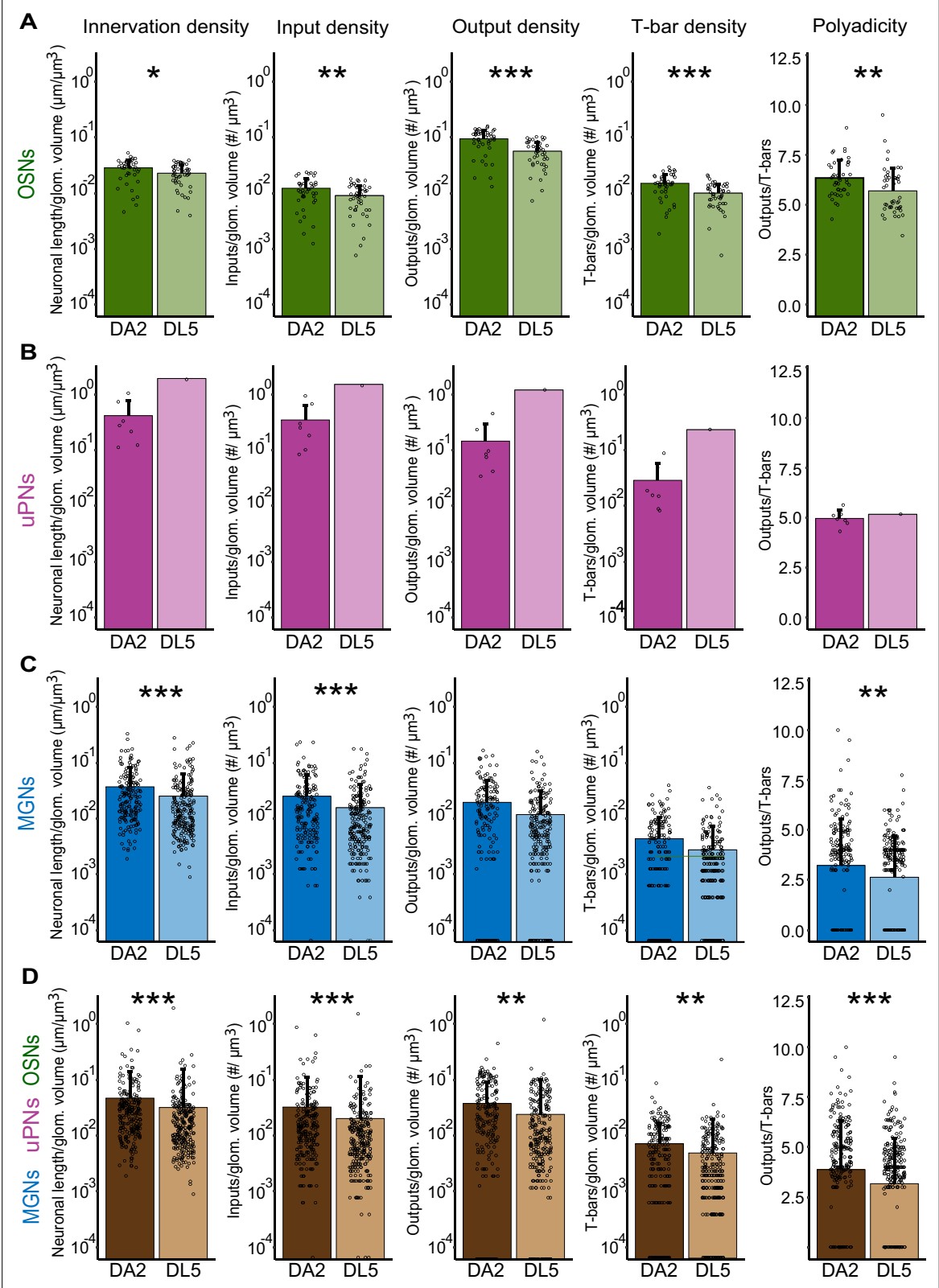

**Figure 3.** Innervation density and synaptic density in DA2 and DL5. Quantification of average glomerular innervation density of olfactory sensory neurons (OSNs) (**A**), uniglomerular projection neurons (uPNs) (**B**), multiglomerular neurons (MGNs) (**C**), and all glomerular neurons combined (**D**) and synaptic density of input sites (inputs), output sites (outputs), T-bars, and average polyadicity. Innervation density: total length (µm) of each neuronal fiber normalized to one µm$^3$ of glomerular (glom.) volume. Synaptic density: number of input sites, output sites, or T-bars for each neuronal fiber normalized

*Figure 3 continued*

to 1 µm³ of glomerular volume. Polyadicity: average number of single output sites per T-bar for each neuronal fiber. Data for DA2 is shown in dark colors and for DL5 in light colors. Number of neurons in DA2: OSNs (green) *n* = 44; uPNs (magenta) *n* = 7; MGNs (blue) *n* = 180; all neurons *n* = 231, in DL5: OSNs *n* = 46; uPN *n* = 1; MGNs *n* = 221; all neurons *n* = 268. Data are presented as mean with standard deviation (error bars). Data points represent single neuron values. Means were compared using either Student's *t*-test (OSNs) or Wilcoxon two-sample test (MGNs and all neurons). uPNs were not compared to the single uPN of the DL5 glomerulus. Significance value: p > 0.05 (not significant, no star), p ≤ 0.05 (*), p ≤ 0.01 (**), p ≤ 0.001 (***). Values are provided at data availability; polyadicity values are listed in *Table 1*, row 14.

The online version of this article includes the following source data and figure supplement(s) for figure 3:

**Source data 1.** Measurements and analysis of neuronal features (olfactory sensory neurons [OSNs], uniglomerular projection neurons [uPNs], and multiglomerular neurons [MGNs]) in DA2 and DL5 (X_DA2_rAl and X_DL5_lAl), visualized in *Figure 3*.

**Figure supplement 1.** Synaptic density along neuronal fibers in DA2 and DL5.

**Figure supplement 1—source data 1.** Measurements and analysis of neuronal features (olfactory sensory neurons [OSNs], uniglomerular projection neurons [uPNs], and multiglomerular neurons [MGNs]) in DA2 and DL5 (X_DA2_rAl and X_DL5_lAl), visualized in *Figure 3—figure supplement 1*.

the DL5 (*Figure 3B*). The total glomerular synaptic input density of all uPNs was higher in DA2 as compared with DL5 (*Table 1*, row 6). On the other hand, the total glomerular synaptic density of the T-bars and output sites was similar in DA2 and DL5 (*Table 1*, rows 7 and 8). In line with these results, the neuronal density of T-bars and output sites was less in the DA2 uPNs compared with the DL5 single uPN, whereas the neuronal density of input sites was similar (*Figure 3—figure supplement 1*; *Table 1*, rows 9 and 10). This resulted in synaptic ratios (T-bars-to-inputs and outputs-to-inputs) in the DL5 uPN that were nearly twice as high as those in the DA2 uPNs (*Table 1*; rows 12 and 13).

We then hypothesized that DA2 would exhibit a lower innervation density of MGNs (mainly LNs) compared to DL5, as previous studies have reported that DA2 is innervated by fewer LNs (*Chou et al., 2010*; *Grabe et al., 2016*). However, we observed the opposite: the innervation density of MGNs was significantly higher in DA2 than in DL5 (*Figure 3C*), with slightly higher total innervation density (*Table 1*, row 5). Interestingly, only the glomerular input density was significantly higher for DA2 MGNs compared to that found in DL5, not the glomerular synaptic density of output sites or of the T-bars (*Figure 3C*). However, the total glomerular synaptic density of input sites, output sites, and T-bars was still higher in DA2 than in DL5 (*Table 1*, rows 6–8). The synaptic densities along the MGN fibers were similar in DA2 and DL5 (*Figure 3—figure supplement 1*).

In summary, the DA2 glomerulus is more densely innervated than DL5 and has a more densely packed neuropil with more synaptic contacts relative to the DL5. The DA2 has a significantly higher innervation density and higher density of T-bars, output and input sites per volume (*Figure 3D*, *Table 1*, rows 5–8). The degree of synapse polyadicity is also significantly higher in DA2 than in DL5 (*Figure 3D*, *Table 1*, row 14) due to a shift to higher polyadicity among OSN (*Figure 3A*) and MGN synapses (*Figure 3C*). OSNs show the strongest shift in polyadicity, with tetrads being the most abundant synapse type in DA2 whereas triads are the most abundant in DL5 OSNs (*Figure 2—figure supplement 1B*).

## Lateralization of OSN glomerular connectivity

In *D. melanogaster*, the majority of olfactory glomeruli receive bilateral OSN input (*Stocker et al., 1983*; *Stocker et al., 1990*; *Vosshall et al., 2000*; *Silbering et al., 2011*) (see scheme in *Figure 4A*). Recent studies have shown that ipsi- and contralateral OSNs are asymmetric in their synaptic connectivity to other neurons in the majority of the glomeruli (*Tobin et al., 2017*; *Schlegel et al., 2021*) and that ipsi- and contralateral OSNs activate uPNs in an asymmetric way (*Gaudry et al., 2013*; *Tobin et al., 2017*). However, not all glomeruli appear to have the same degree of lateralized OSN connectivity (*Schlegel et al., 2021*). At least for one narrowly tuned pheromonal glomerulus (DA1), there is functional evidence that in female flies (but not in males) its uPNs are evenly activated by either ipsi- or contralateral antennal stimulation (*Agarwal and Isacoff, 2011*). We hypothesized that this lack of lateralization could be a feature of other narrowly tuned glomeruli in female flies.

Ipsi- and contralateral OSNs in DA2 and DL5 were identified based on the location and trajectory of their axons (*Figure 4B*). In both glomeruli, ipsilateral OSN terminals were longer than their contralateral counterparts within the VOI, while polyadicity was stronger in contralateral axons. Synaptic density was not consistently higher or lower in ipsilateral OSNs compared to contralateral ones in DA2 and DL5 (*Figure 4—figure supplement 1*).

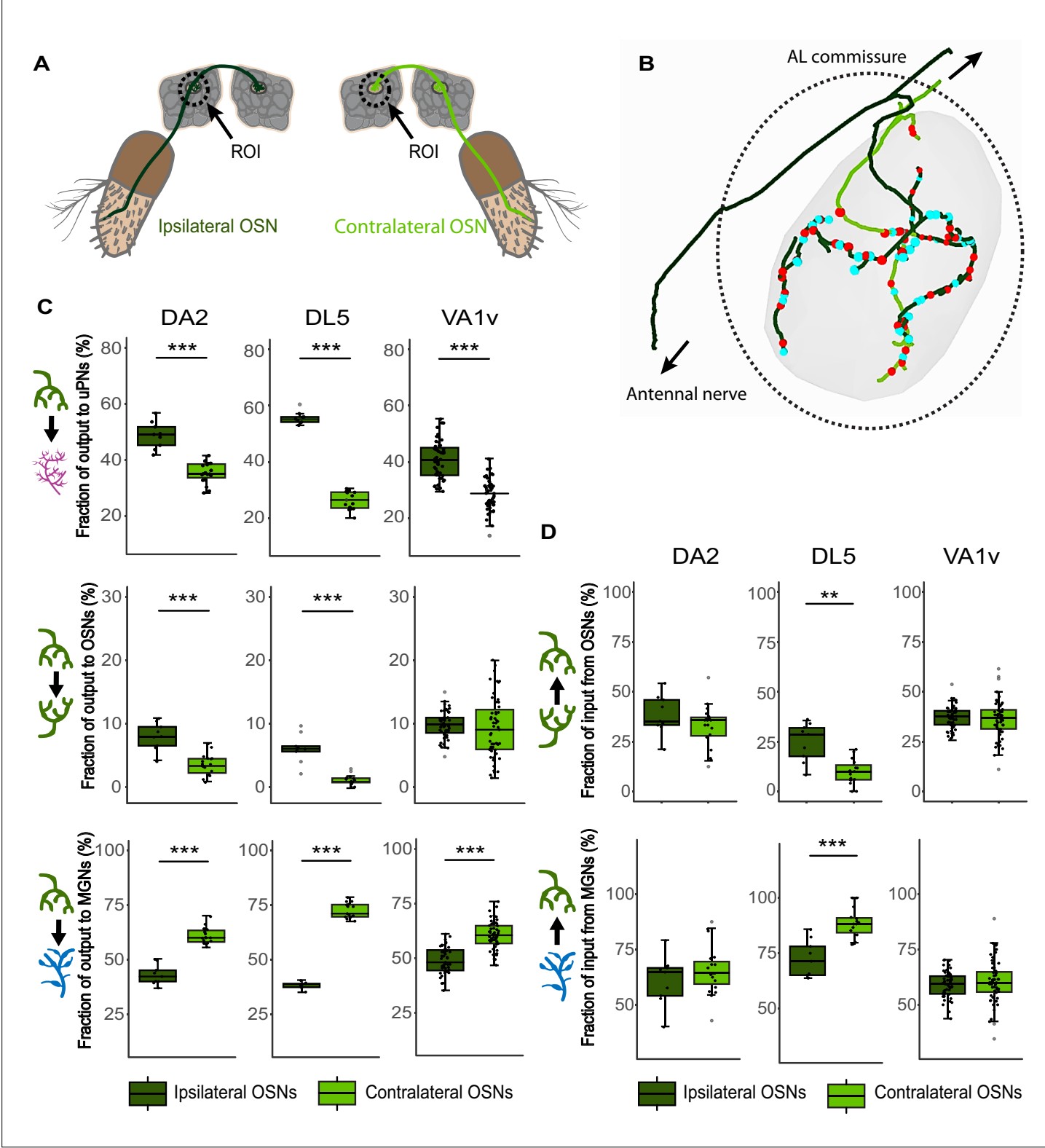

**Figure 4.** Lateralization of olfactory sensory neuron (OSN) terminals in the antennal lobes (ALs). (**A**) Illustration of an ipsilateral (dark green) and a contralateral (light green) OSN with dendrites in the corresponding antennae and their axonal projections to the ipsilateral olfactory glomerulus in the AL (dashed rectangle). (**B**) Exemplary skeleton traces of an ipsilateral (dark green) and a contralateral (light green) OSN terminal inside glomerulus DA2. The ipsilateral OSN axons reach the glomerulus via the ipsilateral antennal nerve (arrow down) and leave the glomerulus toward the AL commissure (arrow up) while OSN axons originating at the contralateral antenna reach the glomerulus via the AL commissure. Red dots: presynaptic sites; blue dots:

*Figure 4 continued on next page*

*Figure 4 continued*

postsynaptic sites. (**C**) Boxplots showing the fraction of synaptic output to uniglomerular projection neurons (uPNs, in magenta), to OSNs (in green), or to multiglomerular neurons (MGNs) (in blue), for the ipsilateral OSNs (dark green boxplot) and contralateral OSNs (light green), respectively, in the DA2, DL5, and VA1v glomeruli. (**D**) Boxplots showing the fraction of synaptic input of the same ipsilateral and contralateral OSNs that they receive from OSNs and MGNs. Connection polarity is indicated by arrows in the schematic neuronal drawings on the left of each plot. Dots represent single values. Means were compared using Student's *t*-test. Significance value: p > 0.05 (not significant, no star), p ≤ 0.01 (**), p ≤ 0.001 (***). Mean and median values are provided at data availability. The data for glomerulus VA1v was extracted from *Horne et al., 2018*.

The online version of this article includes the following source data and figure supplement(s) for figure 4:

**Source data 1.** Analysis of ipsi- and contralateral olfactory sensory neurons (OSNs) in DA2 (X-DA2_rAl).

**Source data 2.** Analysis of ipsi- and contralateral olfactory sensory neurons (OSNs) in DL5 (X-DL5_lAl).

**Source data 3.** Analysis of ipsi- and contralateral olfactory sensory neurons (OSNs) in VA1v extracted from Figure 5 in *Horne et al., 2018*.

**Figure supplement 1.** Properties of ipsi- and contralateral olfactory sensory neurons (OSNs).

We observed that the synaptic output of ipsi- vs. contralateral OSNs was asymmetric, with significant differences in the ipsi- and contralateral OSN output to either uPNs, OSNs, or MGNs (*Figure 4C*, DA2 and DL5). In agreement with previous observations in other glomeruli (*Schlegel et al., 2021*), the output fraction to uPNs and OSNs was greater in ipsilateral OSNs than in contralateral ones (*Figure 4C*, DA2 and DL5). Vice versa, the OSN output to MGNs was greater in the contralateral glomerulus than in the ipsilateral side (*Figure 4C*, DA2 and DL5). However, the differences between the medians and means were smaller in DA2 than in DL5 (*Figure 4C*; differences between means: see data availability).

Our finding of less lateralized connections in the DA2 (*Figure 4C*, DA2 and DL5) was also observed in another narrowly tuned glomerulus (VA1v; *Dweck et al., 2015*) for which connectome data is available (*Horne et al., 2018*). In VA1v, the OSN output to uPNs and MGNs was significantly asymmetric in the same manner as in DA2, that is, with greater ipsilateral OSN output fractions to uPNs and greater contralateral OSN output fraction to MGNs (*Figure 4C*). However, asymmetry in the VA1v OSN output fractions was even less distinct than in DA2 (regarding both the difference between the median and the mean (*Figure 4C*) and data availability). In addition, in VA1v, the OSN output fraction to other OSNs was symmetric (*Figure 4C*).

The OSN input, from either sister OSNs or MGNs, was asymmetric in DL5 but not in the narrowly tuned glomeruli (*Figure 4D*). The inputs from uPNs to ipsi- or contralateral OSNs were not compared due to their low numbers.

In summary, our data add to the knowledge of lateralized connectivity within olfactory glomeruli and support the hypothesis that narrowly tuned glomeruli have a lower degree of lateralization of OSN connectivity compared with broadly tuned glomeruli.

## Glomeruli DA2 and DL5 differ in several features of their circuitry

Next, we asked whether the synaptic circuitries of DA2 and DL5 differ from each other. We counted each synaptic contact (*Figure 5—source data 3 and 4*) and categorized the distinct connection motifs according to the neuron class to which the output and input neuron belonged (*Figure 5A*; *Figure 5—source data 1*). Each connection motif (for example OSN > uPN, i.e., the OSN-to-uPN feedforward connection) was then assessed for its relative synaptic strength, that is, how many synaptic contacts of this particular connection motif were found compared to the total number of synaptic contacts within the respective circuitry (*Figure 5A–D*; see Methods).

We found that neurons from each class made synaptic contacts with each other in DA2 and DL5, as previously reported for other glomeruli (*Berck et al., 2016*; *Rybak et al., 2016b*; *Tobin et al., 2017*; *Horne et al., 2018*; *Schlegel et al., 2021*). In both DA2 and DL5, OSNs provided the strongest relative synaptic output, that is, 49% of all synaptic connections in DA2 and 43% in DL5 were formed by OSNs (*Figure 5B, C*). Thus, even though DA2 and DL5 had similar numbers of OSNs (44 and 46, respectively), those in DA2 provided a stronger circuit output (14% stronger; *Figure 5—source data 1*) than those in DL5 (*Figure 5B, C*). In both glomeruli, the main output partners of OSNs were MGNs and uPNs, that is, 27% of all circuitry connections in DA2 and 24% in DL5 were OSN > MGN connections, while 20% in DA2 and 18% in DL5 were OSN > uPN connections (*Figure 5B, C*). In DA2, interestingly, each of the seven uPNs received input from almost all OSNs and so could maintain a

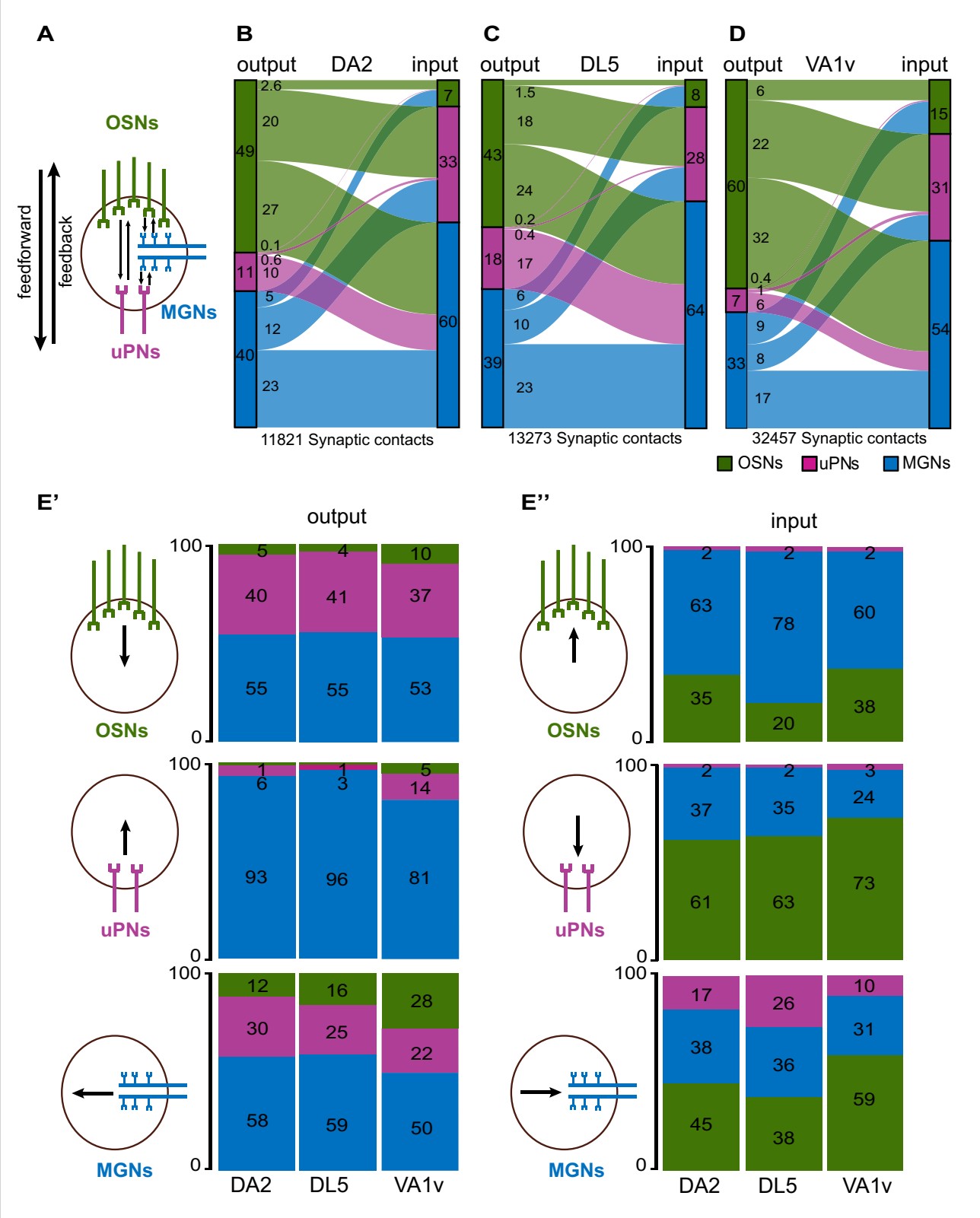

**Figure 5.** Strength of synaptic connections between neuron classes in the circuitry of DA2, DL5, and VA1v. (**A**) Schematic representation of the principal connection motifs among the neuron classes olfactory sensory neurons (OSNs, green), uniglomerular projection neurons (uPNs, magenta), and multiglomerular neurons (MGNs, blue). Synaptic connections directed toward uPNs are defined as feedforward and those directed toward OSNs or from uPNs to MGNs are classified as feedback connections (indicated by arrows). Alluvial diagrams of the glomerular circuitry in DA2 (**B**), DL5 (**C**), and VA1v

*Figure 5 continued on next page*

*Figure 5 continued*

(**D**). Each diagram shows the relative synaptic strength calculated as the proportion of 1:1 single synaptic contact between each neuron class in relation to the total number of synaptic contacts in their respective glomerulus. The synaptic strength between each neuron class, given as a percentage, is indicated by the thickness of the lines. The proportions (as percentage) of output (left side) or input (right side) are illustrated by colored rectangles to the left or right of each alluvial diagram. The total number of synaptic contacts is indicated below the diagrams. Percentages of the relative synaptic strength and synaptic counts are listed in the *Figure 5—source data 1*. (**E**) Stacked bar charts depict output (**E'**) and input (**E''**) fractions (given as percentages) of each neuron class: OSNs (green), uPNs (magenta), MGNs (blue), schematically illustrated next to the bar charts, respectively, to each of the other neuron classes for glomeruli DA2, DL5, and VA1v. Fractions are color-coded according to the neuron class of the respective connecting partner.

The online version of this article includes the following source data for figure 5:

**Source data 1.** Synaptic connectivity and relative differences between DA2, DL5, and VA1v.

**Source data 2.** Calculations of fractional out- and input based on synaptic counts in DA2 (X-DA2_rAl), DL5 (X-DL5_lAl), and VA1v extracted from Figure 5 in *Horne et al., 2018*.

**Source data 3.** Raw data, excel sheet.

**Source data 4.** Raw data, excel sheet.

high degree of convergent signal transmission *Figure 5—source data 3*. In contrast, OSNs received the lowest relative input of all neuron classes in DA2 and DL5 (7% and 8% respectively; *Figure 5B, C*). In line with previous observations in other glomeruli (*Horne et al., 2018*; *Schlegel et al., 2021*), OSNs also made abundant axo-axonic synapses with sister OSNs (2.6% in DA2 and 1.5% in DL5; *Figure 5B, C*). Thus, the relative synaptic strength of the OSN > OSN connection was 70% stronger in DA2 than in DL5 (*Figure 5B, C*; *Figure 5—source data 1*).

The uPNs in both glomeruli had the weakest relative output of all neuron classes within their circuitry, and this was even weaker (38%) in DA2 (*Figure 5B, C*; *Figure 5—source data 1*). In contrast, the relative synaptic input onto uPNs was greater in DA2 than in DL5 (33% vs. 28%, respectively; *Figure 5B, C*; 16% stronger in DA2; *Figure 5—source data 1*), which is in line with our finding that in DA2, the uPNs provide more input sites per unit of glomerular volume than in the DL5 (*Figure 3B, C*). In both glomeruli, the feedback connections from uPNs (depicted in *Figure 5A*) were almost exclusively directed toward MGNs, as previously reported for the broadly tuned DM6 and the narrowly tuned glomerulus VA1v (*Tobin et al., 2017*; *Horne et al., 2018*). However, the relative synaptic strength of the uPN > MGN connection was 40% weaker in DA2 than in DL5 (uPN > MGN: 10% in DA2 and 17% in DL5). Only a few cases of uPN > OSN synaptic connections were observed (a total of 16 in DA2 and 26 in DL5) representing a synaptic strength of 0.1% in DA2 and 0.2% in DL5 (*Figure 5—source data 1*). Finally, uPNs in DA2 also made 71 reciprocal synaptic connections (representing a synaptic strength of 0.6%; *Figure 5—source data 1*; *Figure 5B*), consistent with electrophysiological evidence for reciprocal synaptic interactions between sister uPNs (*Kazama and Wilson, 2009*). The single uPN of the DL5 had 54 dendro-dendritic synapses (representing 0.4% of all DL5 synaptic contacts; *Figure 5C*), which were exclusively autapses, that is, synapses formed by a neuron onto itself. Dendritic uPN autapses also exist in DA2-uPNs, but they were few: we observed only 14 autaptic uPN–uPN connections in DA2, which were mainly located at the two longest uPN dendrites (for further analysis of autapses see next section).

MGNs received the strongest input in both glomeruli (60% of the total input in DA2 and 64% in DL5; *Figure 5B, C*). This is in line with the observation that MGNs provided the majority of all traced neuronal fibers in each glomerulus and had the highest innervation density of all neuron classes; *Table 1*. The relative output strength of MGNs was similar in both glomerular circuits (~40% of the total output in each glomerulus; *Figure 5B, C*). MGNs made many reciprocal synapses to each other, accounting for 23% of all synapses in both glomeruli (*Figure 5B, C*). The relative synaptic strength between MGN > uPN was stronger in DA2 (12%) than DL5 (10%) (*Figure 5B, C*; *Figure 5—source data 1*). The MGN > OSN feedback connection was relatively weak in both glomeruli (5% in DA2 vs. 6% in DL5; *Figure 5B, C*) but weaker (25%) in DA2 than in DL5 (*Figure 5—source data 1*).

We then looked at the fractional output and input of each neuron class (*Figure 5E', E''*). In both glomeruli, OSNs had a similar proportion of their synaptic output onto uPNs (40–41%), onto MGNs (55% in both), and onto sister OSNs (4–5%) (*Figure 5E'*). From the uPNs perspective, over 93–96% of their recurrent synaptic output was directed to MGNs in both DA2 and DL5, and few synapses were directed onto OSNs (~1% of the uPN output; *Figure 5E'*). The uPN > uPN output fraction

of the seven uPNs in DA2 (reciprocal synapses) was twice the uPN output fraction (autaptic) of the single uPN dendrite in DL5 (6% vs. 3%; *Figure 5E'*). MGNs formed synaptic output mainly to other MGNs (58–59% of the total MGN output in DA2 and DL5). Among MGNs, we found also rare cases of autapses. The MGN > uPN output fraction was greater in DA2 (30%) than in DL5 (25%), whereas the MGN > OSN output fraction was smaller in DA2 (12%) than in DL5 (16%; *Figure 5E'*).

Turning to the input fractions of each neuron class, we found that in both glomeruli, OSNs received most of their input from MGNs (>50%). In DA2, the input fraction onto OSNs (MGN > OSN) was smaller than in DL5 (63% vs. 78%; *Figure 5E''*). In contrast, the OSN input fraction from sister OSNs was greater in DA2 (35% vs. 20%; *Figure 5E''*). In both glomeruli, the OSNs received only weak uPN input (2%) (*Figure 5E''*). The input fractions onto the seven uPNs, formed by uPNs, MGNs, and OSNs, in the DA2 and the single uPN in DL5 were similar (*Figure 5E''*). Most uPN input was delivered by OSNs (~62% in both glomeruli) and less from MGNs (~36%). The uPN input fraction from other uPNs in DA2 or the autaptic input from the single uPN in DL5 was small (2%; *Figure 5E''*). In DA2, the MGNs received a smaller fraction of uPN feedback input than in DL5 (17% vs. 26%; *Figure 5E''*) but a greater OSN input fraction (45% vs. 38%; *Figure 5E''*). The fraction of MGN > MGN input was similar in both glomeruli.

To further explore whether the differences found in the circuitry between DA2 and DL5 reported here might represent features characteristic of narrowly tuned glomeruli, we analyzed connectome data from another narrowly tuned glomerulus (VA1v; *Horne et al., 2018*). We calculated the relative synaptic strength between OSNs ($n$ = 107), uPNs ($n$ = 5), and MGNs ($n$ = 74) in the VA1v (*Figure 5D*; *Figure 5—source data 1*). We found that the two narrowly tuned glomeruli shared five circuit features that differentiate them from the broadly tuned glomerulus DL5: First, OSNs in VA1v, as reported above for DA2, displayed a stronger relative feedforward output to uPNs (22%) and to MGNs (32%), which was even larger than in DA2 (*Figure 5D*). The uPNs and MGNs in VA1v received a larger fraction of OSN input than in DL5 (*Figure 5E''*). Second, the OSN > OSN synaptic output was four times stronger (6%) than in DL5 (1.5%; *Figure 5B–D*, *Figure 5—source data 1*). This was also reflected in the OSN output fraction to sister OSNs (10%), which in VA1v was more than twice that of DL5 (4%; *Figure 5E'*) and in the much greater OSN input fraction (38%) to OSNs in the VA1v than in DL5 (20%; *Figure 5E''*). This was in line with previous connectome analysis, showing much stronger OSN input from sister OSNs in narrowly tuned glomeruli, such as DL4, DL3, or DA1 compared to the broadly tuned ones, such as DM4 or DM1 (*Schlegel et al., 2021*). Third, in the VA1v, the uPN > uPN relative synaptic output was more than twice that of the DL5 uPN (mediated by autapses) (1% vs. 0.4% in DL5; *Figure 5D*), which is in accordance with a much greater uPN output fraction to uPNs (14%) in VA1v than in DL5 (3%) (*Figure 5E'*). Forth, VA1v uPNs had fewer feedback synapses onto MGNs than in DL5 (relative synaptic strength of uPN > MGN connection: 6% vs. 17%; *Figure 5C, D*), also reflected in a smaller output fraction from uPNs to MGNs in VA1v than in DL5 (81% vs. 96%; *Figure 5E'*). In agreement, the MGN input fraction from uPNs in VA1v was much smaller than in DL5 (10% vs. 26%; *Figure 5E''*). Fifth, OSNs in VA1v received a smaller MGN input fraction than DL5 OSNs (60% vs. 78%; *Figure 5E''*).

In addition to the shared connectivity differences (either stronger or weaker) between DA2 and VA1v that distinguish them from DL5, two connection motifs were found to be stronger in DA2 and DL5 than in VA1v: The MGN > uPN connection showed a synaptic strength of 12% and 10% in DA2 and DL5 vs. 8% in VA1v (*Figure 5B–D*, *Figure 5—source data 1*). In agreement with this, the MGN output fraction to uPNs (*Figure 5E'*, MGN output) and the MGN input fraction in uPNs was greater in DA2 and DL5 than in VA1v (*Figure 5E''*, uPN input). The relative synaptic strength in MGN > MGN motifs was similar between DA2 and DL5 (23%; *Figure 5B, C*), but weaker in VA1v (17%; *Figure 5D*, *Figure 5—source data 1*). This was also reflected in a smaller MGN output and input fraction from or to MGNs (*Figure 5E', E''*).

In summary, the two narrowly tuned glomerular circuits studied here shared five circuit features when compared with the broadly tuned glomerular circuit (all glomerular circuit features in DA2, DL5, and VA1v are shown in *Figure 6A*). These features were (1) a stronger OSN > uPN and OSN > MGN connection, (2) a much stronger axo-axonic communication between sister OSNs, (3) a stronger dendro-dendritic connection between uPN dendrites, (4) lateralization of OSN connectivity is weaker in narrowly tuned circuits of DA2/Va1v, (5) less feedback from uPNs to MGNs, and (6) less feedback from MGNs to OSNs (*Figure 6B*). These findings, which may have important computational

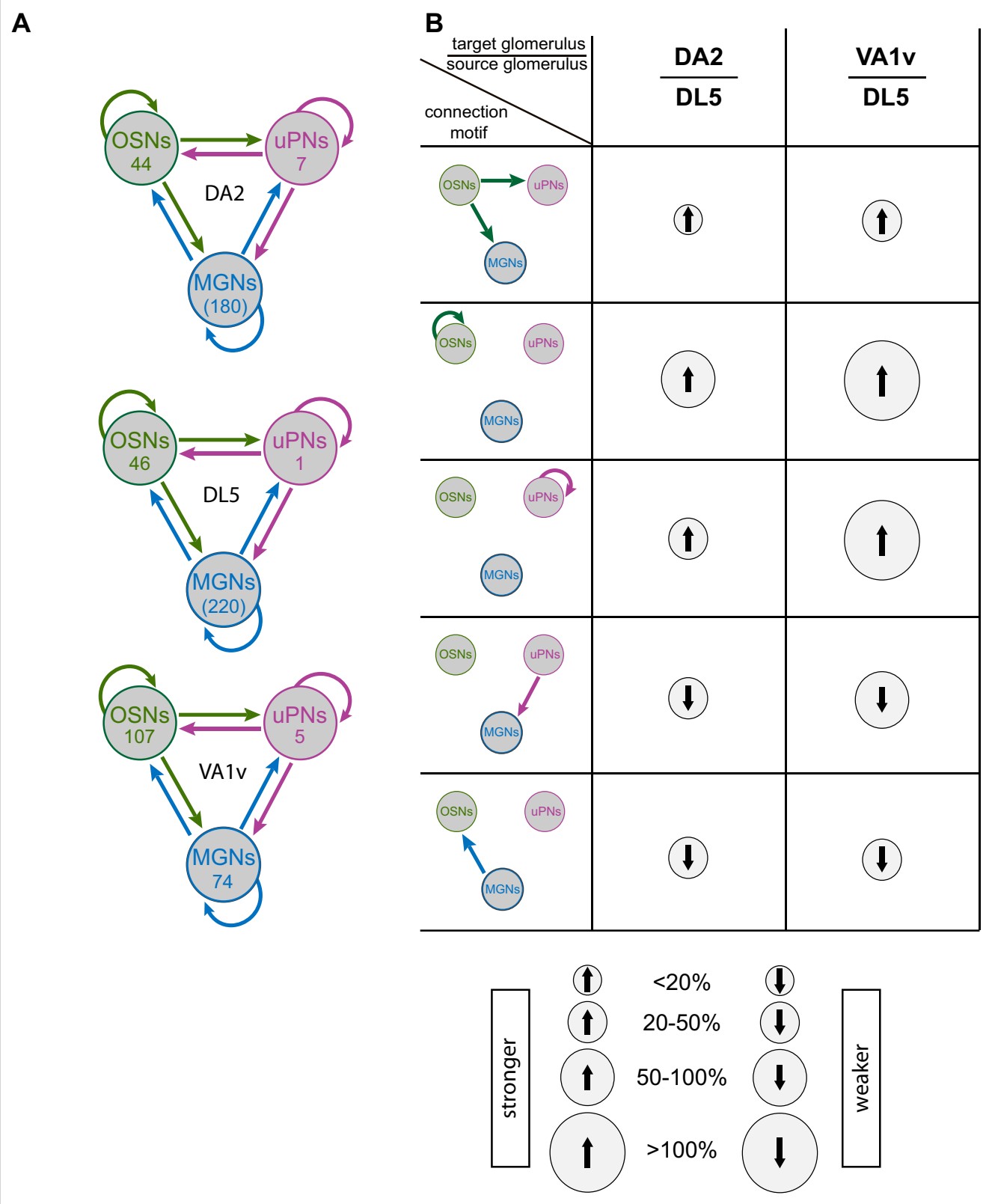

**Figure 6.** Differences in connectivity strength in glomeruli DA2, DL5, and VA1v. (**A**) Schematic representation of synaptic connection motifs (arrows) between olfactory sensory neurons (OSNs, green), uniglomerular projection neurons (uPNs, magenta), and multiglomerular neurons (MGNs, blue) in glomeruli DA2, DL5, and VA1v. The number of neurons of each class – or truncated neuronal fibers (indicated in brackets) – is shown within the corresponding circle. (**B**) Diagrams of connection motifs (left) that are consistently stronger or weaker in DA2 and VA1v compared to DL5. Relative

*Figure 6 continued on next page*

*Figure 6 continued*

differences (expressed as percentages) between DA2 and DL5, and between VA1v and DL5 are illustrated as upward (stronger) or downward (weaker) arrows, with arrow intensity indicating the magnitude of the difference (see legend at the bottom) from the perspective of the target glomerulus (as defined in the table header). All relative difference values are provided in *Figure 5—source data 1*.

implications for glomerular circuits processing a single odorant identity, distinguish them from broadly tuned circuits and are summarized in the graphical representation in Figure 8.

## Autapses in the large DL5 uPN connect distant regions of its dendritic tree

Autapses (synapses made by a neuron upon itself) have seldom been reported in the *Drosophila* central nervous system (*Takemura et al., 2015*; *Horne et al., 2018*). In the DA2 glomerulus, we found few autapses in uPNs and MGNs (*Figures 5C and 7A*). In contrast, within the dendritic tree of the single DL5 uPN, three observers independently identified 54 autaptic connections (see Methods). This represents 3% of the output connections of this neuron and 0.4% of all synaptic contacts in the whole glomerulus (*Figures 5C, E′ and 7A*). We found that these autapses were not distributed evenly along the dendritic tree of the DL5 uPN. Some dendritic branches received several autaptic inputs, whereas others had no autaptic input (*Figure 7A*) and we hypothesized that these autapses could connect distant parts of this very large dendritic tree. We thus analyzed the exact location and distribution of their pre- and postsynaptic sites (*Figure 7A*). We discovered a difference in the distribution of the pre- and postsynaptic elements of DL5 autapses. While their presynaptic T-bars were evenly distributed at basal (Strahler order: 5) and at distal regions (Strahler order: 1–4), 95% of their postsynaptic sites were located in the most distal region (Strahler order 2–1; *Figure 7B, C*). We also calculated the geodesic distance (i.e., along-the-arbor distance) from pre- and postsynaptic sites to the basal root node, which is the node point where the DL5 uPN enters the glomerulus and is equivalent to the closest point to the soma in our reconstruction. The geodesic distance to the basal root node from the presynaptic site was significantly shorter than for postsynaptic sites (*Figure 7—figure supplement 1B*). The pre- and postsynaptic sites of each autapse were either close to each other along the dendritic tree or distant from each other (see examples in the dendrogram depicted in *Figure 7D*). Thus, the geodesic distance between pre- and postsynaptic sites (see scheme in *Figure 7E*), as well as the number of branching points between pre- and postsynaptic partners, was bimodally distributed (*Figure 7F, G*). Autapses that connected distant dendritic branches were more frequent than those that connected close dendritic branches (*Figure 7E–G*). In summary, we found abundant autapses within the uPN dendrite of DL5, and they were unevenly distributed, with many output sites located in a few sub-branches connecting distal dendritic regions.

## Discussion

We hypothesized that specialized, narrowly tuned olfactory glomeruli differ in their ultrastructure and microcircuitry from broadly tuned glomeruli. By comparing data from dense reconstructions of two narrowly tuned olfactory glomeruli with that of a broadly tuned glomerulus in *D. melanogaster*, we identified prominent features of narrowly tuned glomeruli related to synaptic composition, lateralization of sensory input, and synaptic circuitry.

### Glomerular circuit analysis: a correlative approach

The small size of olfactory glomeruli in *Drosophila* gave us the opportunity to reconstruct and analyze the dense connectome of entire glomeruli with volume-based EM in a reasonable time period. Here, we developed a correlative workflow that combines transgenic neuron labeling with near-infra-red-laser branding for precise volume targeting. We then used FIB-SEM (*Bishop et al., 2011*) to resolve glomerular networks at the synaptic level. A similar procedure was used recently to investigate single cellular organelles (*Ronchi et al., 2021*). An advantage of this approach is that it facilitates localization of the VOI with high precision and consequently limits to a minimum the volume to be scanned and reconstructed. At the same time, the limitation in volume is a drawback of our workflow, as it was impossible to reconstruct neurons back to their soma. This fact prevented the identification of individual neurons as in other connectome studies (*Berck et al., 2016*; *Eichler et al., 2017*; *Horne*

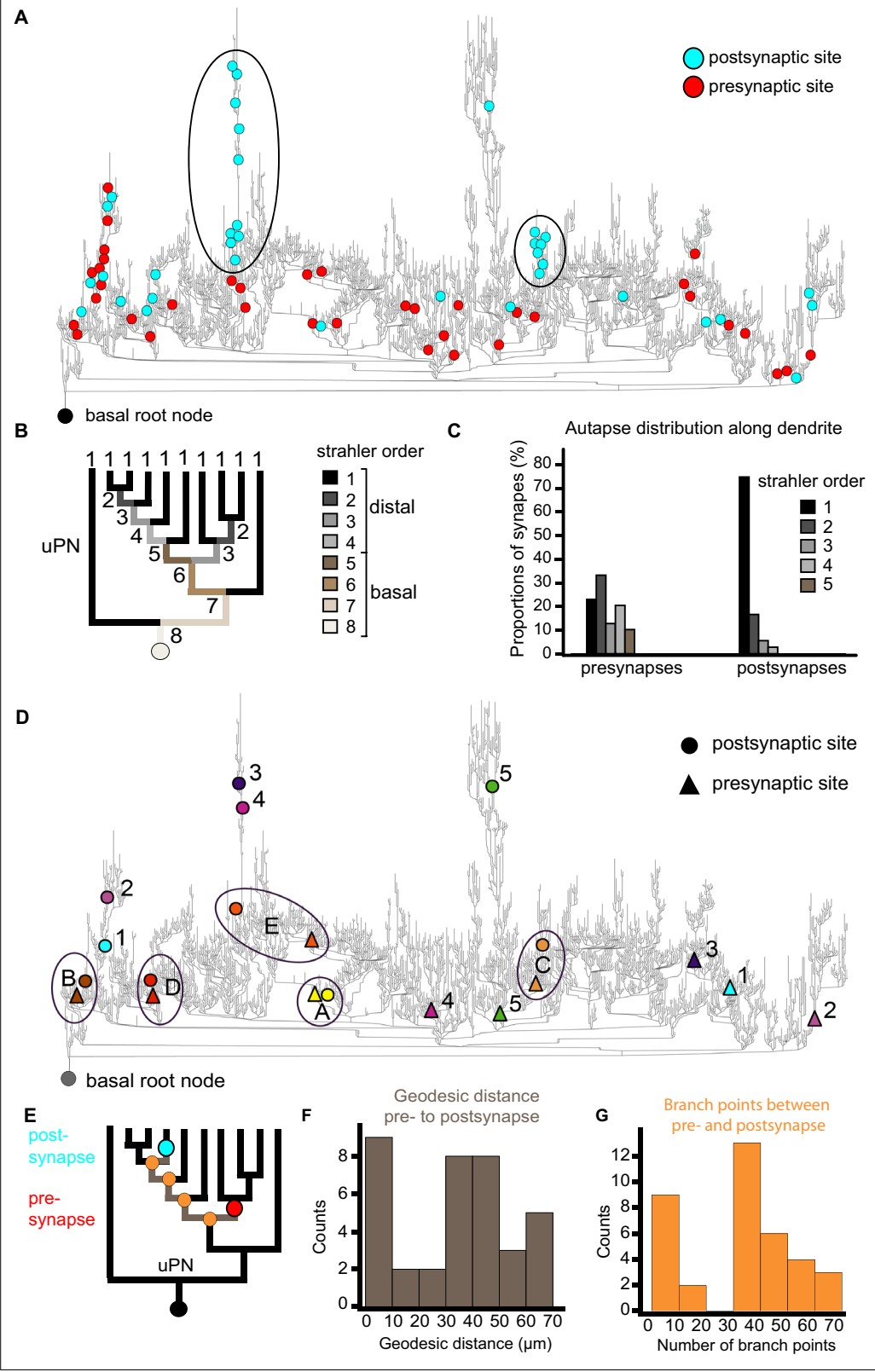

**Figure 7.** Distribution of pre- and postsynaptic partners of autapses in the uniglomerular projection neuron (uPN) dendrite of the DL5. (**A**) Distribution of autaptic presynaptic (red dots) and postsynaptic (cyan dots) sites mapped onto a dendrogram of the single uPN dendrite in glomerulus DL5. The basal root node (black dot) marks the entry site of the uPN dendrite into the glomerulus (i.e., the point closest to its soma). Clusters of autaptic input sites

*Figure 7 continued on next page*

*Figure 7 continued*

along specific branches are encircled. (**B**) Simplified dendrogram of the uPN illustrating distinct Strahler orders, with distal branches (orders 1–4) and at basal branches (orders 5–8); see legend on the right. (**C**) Distribution of autaptic presynaptic (left) and postsynaptic (right) input sites along the dendrite, shown as proportions at each corresponding Strahler order (color-coded). Note that autaptic postsynaptic sites are located almost exclusively at the most distal dendritic branches. (**D**) Dendrogram of the DL5-uPN showing the locations of presynaptic sites (triangles) and postsynaptic sites (circles) for selected autapses (color-matched pairs). Autapses with large geodesic distance between their components are labeled with numbers; those with short distances are encircled and labeled with letters. (**E**) Schemes of the dendrogram illustrating the location of the presynaptic (red dot) and postsynaptic (cyan dot) sites of individual autapses, the geodesic distance between them (measured along the dendrite in µm), and the number of branching points (orange dots) separating the pre- and postsynaptic components. (**F**) Histogram showing the number of autapses grouped by geodesic distance between their pre- and postsynaptic sites (as illustrated in **E**). (**G**) The number of autapses categorized by the number of branch points between their pre- and postsynapses along the uPN dendrite (as illustrated in **E**).

The online version of this article includes the following figure supplement(s) for figure 7:

**Figure supplement 1.** Distribution of synapses and autapses along the DL5 uniglomerular projection neuron (uPN) dendrite in DL5 (**A**).

---

*et al., 2018*; *Zheng et al., 2018*; *Bates et al., 2020*; *Scheffer et al., 2020*; *Xu et al., 2020*; *Schlegel et al., 2021*).

We provide data on innervation and synapse density of OSNs, uPNs, and MGNs in the *Drosophila* AL. We observed a higher innervation density of all neuron types – primarily uPNs and MGNs – as well as higher density of synaptic contacts along OSN terminals in the narrowly tuned DA2 compared with DL5. These results suggest that narrowly tuned glomeruli have a more densely packed neuropil, with more numerous synaptic connections in the feedforward motifs OSN > uPN and OSN > MGN. Overall, our observations on synapse density were comparable with previous reports (*Mosca and Luo, 2014*; *Rybak et al., 2016b*; *Horne et al., 2018*).

## Specific features of narrowly tuned glomerular circuits

Our analysis ultimately revealed six circuit features shared by the narrowly tuned glomeruli DA2 and VA1v (with VA1v data analyzed from *Horne et al., 2018*) that distinguish them from the circuitry of the broadly tuned DL5 and may represent adaptations specific to such dedicated glomerular circuits (see summary in *Figure 8*). However, future studies analyzing the precise number of synaptic connections across a larger number of individuals are needed to assess intra- and inter-animal variability (*Schlegel et al., 2024*). In addition, combining these anatomical observations with physiological studies and computational modeling is essential to evaluate the functional relevance of the observed connectivity differences and to test our hypothesis regarding generic circuit-level distinctions between narrowly and broadly tuned olfactory glomeruli (*Figure 8*).

## The OSN > uPN feedforward connection is stronger in narrowly tuned glomeruli

Presynaptic OSN terminals provide the major input to uPNs in insect olfactory glomeruli (*Hansson and Anton, 2000*; *Chen and Shepherd, 2005*; *Kazama and Wilson, 2008*; *Lei et al., 2010*; *Tobin et al., 2017*; *Horne et al., 2018*; *Rybak and Hansson, 2018*; *Schlegel et al., 2021*). Here, we showed that this connection is stronger in DA2 and VA1v than in DL5 (*Figures 5 and 6*). A strong OSNs > uPN synaptic connection will drive non-linear signal amplification, which improves signal detection at low odor concentrations (*Ng et al., 2002*; *Bhandawat et al., 2007*; *Kazama and Wilson, 2008*; *Masse et al., 2009*). A larger number of synapses of this type could be an adaptation to improve this amplification effect, as shown by artificial increase of synaptic sites in the AL (*Acebes and Ferrús, 2001*) and in lateral horn dendrites (*Liu et al., 2022*).

Each of the seven uPNs in DA2 received convergent synaptic input from almost all DA2-OSNs. This is in agreement with reports on the narrowly tuned glomeruli DA1 and VA1v (*Agarwal and Isacoff, 2011*; *Jeanne and Wilson, 2015*; *Horne et al., 2018*) and for broadly tuned glomeruli (*Vosshall et al., 2000*; *Chen and Shepherd, 2005*; *Kazama and Wilson, 2009*; *Masse et al., 2009*; *Tobin et al., 2017*). High OSN > uPN convergence is the main driver of highly correlated activity among uPNs in

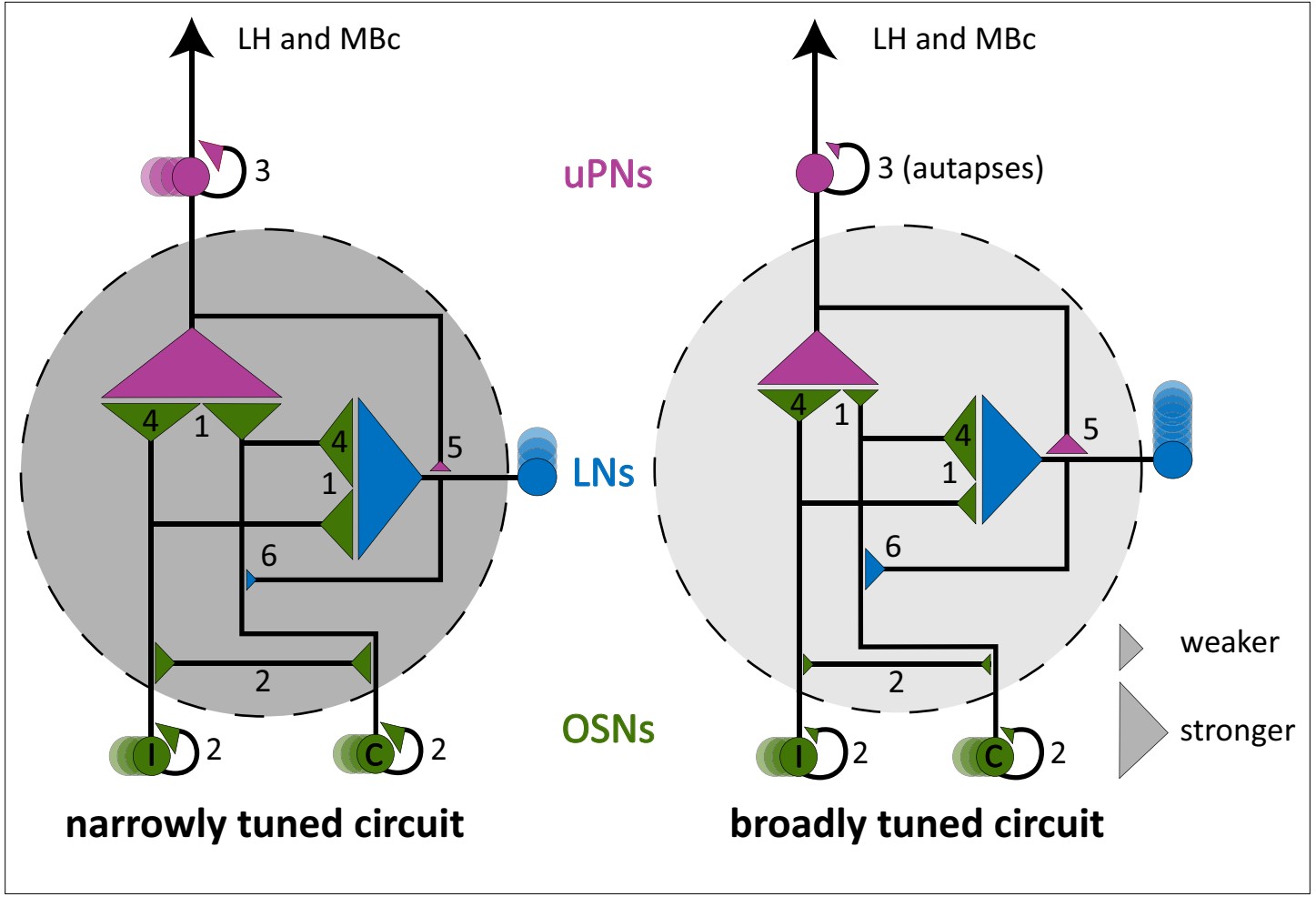

**Figure 8.** Graphic summary illustrating general differences between narrowly and broadly tuned olfactory circuits. Graphic summary of key circuit features that distinguish the two narrowly tuned olfactory glomeruli studied here (dark gray circle) from a broadly tuned on (light gray circle). Olfactory glomeruli are the first relay station where olfactory information is processed before being transmitted to higher brain centers, including the lateral horn (LH) and mushroom body calyces (MBc). The model is based on a comparative analysis of the narrowly tuned circuits of DA2 and VA1v and the broadly tuned circuit of DL5. Circuit components include uniglomerular projection neurons (uPNs, magenta), local interneurons (LNs, blue), and olfactory sensory neurons (OSNs, green) from both ipsilateral and contralateral brain hemispheres. In the narrowly tuned circuits, multiple uPNs are present, whereas in the broadly tuned circuit LNs are more numerous, as indicated by the number of circles. Connectivity strength between neuronal types is inferred from synaptic counts (1:1 presynaptic–postsynaptic sites) and is represented by the size of the connecting triangles. Differences in neuron number are indicated by the number of circles. In narrowly tuned circuits, OSNs exhibit stronger synaptic output (1) and form stronger reciprocal connections with sister OSNs (2) and uPNs (3). The lateralization of OSN connectivity is reduced compared to broadly tuned circuits, where ipsilateral output to uPNs and contralateral output to the LNs dominates (4). Feedback synapses from uPNs to LNs (5) and from LNs to OSNs (6) are weaker in narrowly tuned circuits. In contrast, the broadly tuned circuit DL5 exhibits weaker OSN output and stronger lateralization of OSN inputs. Additionally, autapses are observed in the single uPN of DL5, whereas reciprocal uPN connections are a feature of narrowly tuned circuits. The model attributes to LNs the features quantified for a larger group of neurons (multiglomerular neurons, MGNs), of which the great majority are LNs, as discussed in the main text.

pheromone coding glomeruli in flies as well as moths (*Kazama and Wilson, 2009Kazama and Wilson, 2009*; *Rospars et al., 2014*). High convergence in the lateral horn improves signal transmission from uPNs to lateral horn neurons without sacrificing speed (*Jeanne and Wilson, 2015*; *Huoviala et al., 2020*). In the mushroom body calyces, however, the high degree of convergence is only pursued for DA2 uPNs, which converge onto few Kenyon cells, whereas VA1v uPNs synapse randomly onto many dispersed Kenyon cells (*Caron et al., 2013*; *Li et al., 2020b*), indicating diverse signal integration in the mushroom body.

From our study, we hypothesize that in narrowly tuned glomerular circuits, which have more uPNs, the maintained strong OSN > uPN convergence, improves signal transmission accuracy. Second, a

stronger OSN > uPN connection might compensate for the lack of OSN > uPN signal transmission sites in the case of odorants activating OSNs in a single glomerulus.

## Reciprocal connections between sister OSNs and sister uPNs are stronger in narrowly tuned glomeruli

The reciprocal OSN–OSN synapse is generally stronger in narrowly tuned glomeruli DA1, DL3, and DL4, compared with broadly tuned glomeruli DL5, DM1, DM4, and DM6 (*Suh et al., 2004*; *Knaden et al., 2012*; *Dweck et al., 2015*; *Ebrahim et al., 2015*; *Grabe et al., 2016*; *Seki et al., 2017*; *Tobin et al., 2017*; *Schlegel et al., 2021*). A high degree of axo-axonic synapses between sister OSNs was also found in VA1v (*Horne et al., 2018*; *Schlegel et al., 2021*) and DA2 but not in the DL5 (this study). Hence, we suggest that a strong OSN–OSN connection is a characteristic feature of the synaptic circuitry of narrowly tuned olfactory glomeruli. Axo-axonic connections have also been reported between gustatory and mechanosensory neurons in *Drosophila* larvae (*Miroschnikow et al., 2018*) and in the olfactory epithelium and the olfactory bulb of vertebrates (*Hirata, 1964*; *Shepherd et al., 2021*). In vertebrates, axo-axonic synapses between excitatory sensory neurons are involved in correlated transmitter release (*Cover and Mathur, 2021*), reminiscent of correlated uPN activity due to reciprocal synaptic and electric coupling in the *Drosophila* AL and LH (*Kazama and Wilson, 2009*; *Huoviala et al., 2020*). A strong OSN–OSN connection also has the potential to increase the correlation of OSN spiking events and therefore facilitate a robust OSN signal (*De la Rocha et al., 2007*).

Reciprocal dendro-dendritic synapses between sister uPNs are reported here for the DA2 and have been reported previously also for glomeruli DM6, DM4, VA7, and VA1v (*Kazama and Wilson, 2009*; *Rybak et al., 2016b*; *Tobin et al., 2017*; *Horne et al., 2018*). These types of synapses enhance uPN signal correlation (*Kazama and Wilson, 2009*), as reported for mitral and tufted cells of the vertebrate olfactory bulb, the circuit equivalent to PNs of insect ALs (*Christie et al., 2005*; *McTavish et al., 2012*; *Shepherd et al., 2021*). In *Drosophila*, multiple uPNs could induce correlated PN depolarization events, which improve the signal-to-noise ratio of PN signal transmission (*Chen and Shepherd, 2005*; *Kazama and Wilson, 2009*; *Jeanne and Wilson, 2015*).

In summary, our data give evidence that reciprocal OSN–OSN and uPN–uPN connections are a prominent feature of the synaptic circuit of narrowly tuned glomeruli. We suggest that those reciprocal OSN–OSN and uPN–uPN connections support correlation of neuronal activity and therefore boost signal-induced depolarization events. This will, in turn, enhance the signal-to-noise ratio (accuracy) and transmission probability of weak and/or irregular odorant input, increasing processing speed.

## Less lateralization in the OSN bilateral connectivity in narrowly tuned glomeruli

In *Drosophila*, most OSN axons project bilaterally and form synapses in their corresponding glomerulus on both the left and right brain hemispheres (*Stocker et al., 1990*; *Vosshall et al., 2000*; *Couto et al., 2005*; *Kazama and Wilson, 2009*; *Silbering et al., 2011*; *Tobin et al., 2017*; *Schlegel et al., 2021*). This is rarely observed in other insects and absent in vertebrates (*Stocker et al., 1983*; *Masson and Mustaparta, 1990*; *Galizia et al., 1998*; *Hansson and Anton, 2000*; *Anton et al., 2003*; *Parthasarathy and Bhalla, 2013*; *Dalal et al., 2020*). In the mammalian olfactory system, bilateral comparison of olfactory input only occurs in higher brain centers (*Dalal et al., 2020*). In flies, bilateral sensory input enables them to discriminate odor sources of different spatial origin through bilateral comparison of olfactory stimulation (*Borst and Heisenberg, 1982*; *Duistermars et al., 2009*; *Gaudry et al., 2013*; *Mohamed et al., 2019a*; *Taisz et al., 2023*). Asymmetric OSN connectivity, shown for many olfactory OSNs (*Tobin et al., 2017*; *Schlegel et al., 2021*), seems to be the origin of a bilateral contrast in the uPN response (*Agarwal and Isacoff, 2011*; *Gaudry et al., 2013*; *Tobin et al., 2017*; *Taisz et al., 2023*), and is most likely the key to precise odor source localization (*Taisz et al., 2023*). Bilateral comparison is also used in the lateral horn (a higher olfactory brain center in *Drosophila*) for odorant position coding (*Mohamed et al., 2019a*). However, not all glomeruli are similar in the magnitude of bilateral asymmetry with respect to their OSN connectivity (*Schlegel et al., 2021*) or their uPN responses (*Agarwal and Isacoff, 2011*).

In agreement with observations in most of the olfactory glomeruli (*Schlegel et al., 2021*; *Taisz et al., 2023*), we found that glomeruli DL5, DA2, and VA1v have ipsilaterally asymmetric OSN synaptic output to excitatory uPNs and sister OSNs and contralaterally an enhanced OSN > MGN output

(*Figure 4*). We believe that, in agreement with a recent study, these asymmetric connections determine a strong left–right contrast in the uPN response, akin to a 'winner-takes-all' principle (*Taisz et al., 2023*).

We also observed that the degree of bilateral OSN asymmetry in DA2 and VA1v was much weaker than in DL5 (*Figure 4*). Weakly lateralized OSN connectivity is perhaps insufficient to induce an adequate bilateral contrast necessary for odor source localization. Recent work supports this idea by showing the importance of the interplay of asymmetric OSN signaling and LN inhibition to enhance the bilateral contrast of uPN activity and to facilitate navigation (*Taisz et al., 2023*).

Why do these narrowly tuned glomeruli have weaker bilateral contrast than broadly tuned glomeruli? The answer could lie in the ecological significance of the individual odorants. Geosmin, encoded by glomerulus DA2 (*Stensmyr et al., 2012*), and the pheromone methyl laurate, encoded by glomerulus VA1v (*Dweck et al., 2015*), act at short distances, mainly when the fly is walking and not flying. Perhaps the behavioral response to geosmin or methyl laurate does not need a precise odor source location. On the other hand, food odor detection at a distance, which happens mainly at flying conditions, needs continuous processing of odor position and body alignment to navigate toward the odor source (*Thoma et al., 2015*; *Demir et al., 2020*). The bilateral OSN projection onto uPNs in DA2 and VA1v potentially has a distinct function other than odor position coding and could, via the enhancement of the effect of convergence of OSN > uPN signal transmission, enhance odor signal amplification (*Bhandawat et al., 2007*; *Kazama and Wilson, 2009*; *Masse et al., 2009*; *Jeanne and Wilson, 2015*).

## Distinct synaptic integration of local modulatory neurons in narrowly tuned glomeruli

MGNs are composed of mPNs that project directly to the LH (*Jefferis et al., 2007*; *Strutz et al., 2014*; *Bates et al., 2020*) and inhibitory and excitatory LNs that interconnect the AL glomeruli (*Masse et al., 2009*; *Okada et al., 2009*; *Chou et al., 2010*; *Seki et al., 2010*; *Liu and Wilson, 2013*). Since LNs are the most numerous and broadly arborizing of the multiglomerular cell types in the AL (*Chou et al., 2010*; *Lin et al., 2012*), we focus our discussion on these neurons. A quantitative analysis of MGN connectivity in the VA1v glomerulus – where LNs and mPNs were classified separately – supports this focus. It shows that 84% of the MGN output originates from LNs, and 57% of MGN input is received by LNs. When excluding the major OSN > mPN and OSN > LN input, the majority of remaining MGN input is from LNs (73%) data source: (*Horne et al., 2018*). Multiglomerular LNs play a crucial role in modulating OSN-to-uPN signal transmission within the olfactory glomeruli (*Masse et al., 2009*; *Chou et al., 2010*; *Seki et al., 2010*; *Galizia, 2014*; *Szyszka and Galizia, 2015*).

Previous observations have shown that glomeruli DA2 and VA1v have a lower number of innervating LNs (*Chou et al., 2010*; *Grabe et al., 2016*) and receive less global interglomerular LN inhibition than broadly tuned glomeruli (*Hong and Wilson, 2015*). We therefore assumed that DA2 or VA1v would have a lower LN innervation density and less LN synaptic integration in their circuitry. However, we did not observe a general lower synaptic integration in DA2 (*Figure 5*) and found a greater MGN innervation density and a higher density of input sites than in DL5. VA1v MGNs, on the other hand, received less synaptic input and provided less output in its glomerular circuit than MGNs in DL5.

Taking a closer look at particular synaptic connection motifs of MGNs, we saw that narrowly tuned glomeruli had a relatively weak uPN > MGN feedback (*Figure 6*, see also *Figure 8*). uPN feedback onto LNs and their reciprocal connection (LN > uPN) were reported in *Drosophila* and other insects, such as honey bees, cockroaches, and moths, but their function is still poorly understood (*Boeckh and Tolbert, 1993*; *Sun et al., 1997*; *Sachse and Galizia, 2002*). In the honey bee, reciprocal dendro-dendritic synapses between excitatory and inhibitory neurons enhance signal contrast and the reliability of true signal representations throughout the AL (*Yokoi et al., 1995*; *Sachse and Galizia, 2002*). Here we could not differentiate the LN types involved in the uPN > MGN synaptic motif. However, the prevailing uPN > LN synapses involve mainly widespread panglomerular LNs in the adult (*Horne et al., 2018*) and larval AL (*Berck et al., 2016*), which are important for combinatorial coding (*Galizia, 2014*; *Sachse and Hansson, 2016*). Thus, weaker uPN > MGN feedback in the narrowly tuned DA2 and VA1v circuits might be a compensatory mechanism to lower the computational demand of interglomerular communication for odor identity coding.

We also observed that OSNs received less MGN feedback in the narrowly tuned DA2 and VA1v than in the DL5, suggesting that the OSNs in DA2 and VA1v receive relatively weak presynaptic input. Pan-glomerular GABAergic LNs constitute the main input to OSN presynaptic sites (*Berck et al., 2016*; *Schlegel et al., 2021*). These inhibitory LNs are prime candidates for mediating presynaptic inhibition onto OSNs (*Wilson and Laurent, 2005*; *Olsen and Wilson, 2008*; *Chou et al., 2010*; *Olsen et al., 2010*) and thought to drive balanced glomerular gain control. They play a key role in odor identity coding by modulating both the balance and dynamics of incoming and varying odor intensities (*Olsen and Wilson, 2008*; *Root et al., 2008*; *Silbering et al., 2008*; *Asahina et al., 2009*; *Wang, 2012*; *Galizia, 2014*; *Hong and Wilson, 2015*; *Szyszka and Galizia, 2015*; *Sachse and Hansson, 2016*). Our data support these observations and provide an argument for why narrowly tuned OSNs receive much lower inhibition during AL stimulation with odorants activating other OSN populations (*Hong and Wilson, 2015*). Even though DA2 and VA1v might receive less interglomerular inhibition, their OSN > MGN output is still strong, in agreement with studies showing that throughout the AL, global lateral inhibition mediated by LNs scales with general OSN activation (*Olsen and Wilson, 2008*; *Hong and Wilson, 2015*).

In summary, narrowly tuned circuits are probably influenced more strongly by intraglomerular than by interglomerular modulation. Narrowly tuned circuits perhaps have greater computational capacities in intraglomerular modulation of signal transmission, which could be important, for example, for PN fine-tuning and response adjustment (*Ng et al., 2002*; *Assisi et al., 2012*).

Above, we discussed putative generic features of narrowly tuned glomerular circuits. Besides these circuit features, we found a strong MGN > MGN connection in the aversive glomerular circuits DA2 and DL5 in contrast to a much weaker MGN > MGN connection in the attractive glomerulus VA1v (*Knaden et al., 2012*; *Stensmyr et al., 2012*; *Knaden and Hansson, 2014*; *Dweck et al., 2015*; *Mohamed et al., 2019b*). Why do aversive olfactory circuits have a stronger MGN > MGN connection than attractive circuits? In the larval *Drosophila* AL, reciprocal LN > LN synapses induce disinhibition induced by a strong connection between the pan-glomerular LNs and a bilateral projecting LN, the Keystone LN, which synapses strongly onto pan-glomerular LNs and selectively onto OSNs, which are activated by attractive food odors. This is thought to be a key feature to switch from homogeneous to heterogeneous presynaptic inhibition and therefore to a selective gain control enhancing contrast between attractive and aversive odor activation (*Berck et al., 2016*). Such balanced inhibitory systems could also be present in the adult *Drosophila* AL, reflected in the strong LN > LN connection in DA2 and DL5. Disinhibition of interglomerular presynaptic inhibition in aversive glomeruli circuits might be important for the fly to stay vigilant to aversive odors, while perceiving attractive cues, for example during feeding conditions so that a fast switch in behavior can be initiated if necessary.

## Autaptic connection within the dendritic tree of a single uPN

We observed autapses along the large dendritic tree of the single DL5-uPN. To our knowledge, this is the first report of bulk dendro-dendritic autapses in the *Drosophila* olfactory system, indicating a cell-type specific occurrence of autapses in the DL5-uPN as reported for other cell types in the optic lobe (*Takemura et al., 2015*). Autapses are also reported to be present at different frequencies in different types of neurons in the mammalian brain (*Van der Loos and Glaser, 1972*; *Tamás et al., 1997*; *Bekkers, 1998*; *Bacci and Huguenard, 2006*; *Ikeda and Bekkers, 2006*; *Bekkers, 2009*; *Saada et al., 2009*). In *Drosophila,* most uPNs are cholinergic (*Yasuyama and Salvaterra, 1999*; *Yasuyama et al., 2003*; *Kazama and Wilson, 2008*; *Tanaka et al., 2012*; *Croset et al., 2018*) and the DL5-uPN autapses reported here might activate either nicotinic or muscarinic acetylcholine postsynaptic receptors. Muscarinic acetylcholine receptors have an inhibitory effect in the Kenyon cells of the mushroom body (*Bielopolski et al., 2019*), but mediate excitation in the AL (*Rozenfeld et al., 2019*).

What could be the function of these autaptic feedback loops within the DL5-uPN dendritic tree? Recent studies in vertebrates show that excitatory autapses enhance neuron bursting and excitability (*Guo et al., 2016*; *Wiles et al., 2017*; *Yin et al., 2018*). Autaptic inhibitory connections have been implicated in circuit synchronization, spike-timing precision, self-stabilization of neuronal circuits, and feedback inhibition (Ikeda and Bekkers, 2006; *Van der Loos and Glaser, 1972*; *Tamás et al., 1997*; *Bekkers, 1998*; *Bacci and Huguenard, 2006*; *Saada et al., 2009*).

Autapses in the DL5 uPN form mainly long-distance feedback loops, connecting distinct dendritic subtrees and the basal dendrite region (closer to the soma) with distal branches. This spatial segregation

is similar to the distribution of non-autaptic pre- and postsynaptic sites in *Drosophila* uPNs, where presynaptic sites are located more frequently at basal dendrites than postsynapses (*Rybak et al., 2016b*) and in other insects, such as *Periplaneta americana* and moths (*Malun, 1991*; *Sun et al., 1997*; *Lei et al., 2010*). Dendro-dendritic autaptic feedback loops connecting basal to distal branches and distinct dendritic subtrees of a large dendritic tree might facilitate activity correlation between distant dendritic subunits, as described for non-autaptic, reciprocal uPN > uPN connections (*Kazama and Wilson, 2009*). This could be important in a large compartmentalized dendrite that receives inhomogeneous excitation by several OSNs at distinct dendritic sites, in order to enhance synchronized depolarization events along the dendrite, supporting signal integration (*Graubard et al., 1980*; *Tran-Van-Minh et al., 2015*). Clustered autapses could mediate local signal input amplification for distinct dendritic subunits (*Kumar et al., 2018*; *Liu et al., 2022*). Autaptic contacts, finally, could be able to shift the uPN membrane depolarization toward the spiking threshold and enhance the firing probability during activation.

In conclusion, we provide a comprehensive comparative analysis of the ultrastructure and synaptic circuitry of two functionally diverse olfactory glomeruli with distinct computational demands, processing either single odorant information in a dedicated olfactory pathway (DA2) or input regarding several odorants and taking part in combinatory coding across distributed glomeruli (DL5). Our work provides an opportunity to gain insight into variations in network architecture and provides fundamental knowledge for future understanding of glomerular processing. By comparing our data with those from another narrowly tuned glomerulus (VA1v), we distilled prominent circuit features that suggest that narrowly tuned glomerular circuits encode odor signals with a weaker left–right contrast, improved accuracy, stronger signal amplification, and stronger intraglomerular signal modulation relative to broadly tuned glomeruli. Our findings reveal the existence of autapses in olfactory glomeruli and indicate that dendro-dendritic autapses play an important role in dendritic signal integration.

# Materials and methods

## Key resources table

| Reagent type (species) or resource | Designation | Source or reference | Identifiers | Additional information |
|---|---|---|---|---|
| Genetic reagent (*D. melanogaster*) | Orco-GAL4; UAS-GCaMP6s | *Vosshall et al., 2000* | | https://bdsc.indiana.edu/ |
| Software, algorithm | R | | RRID:SCR_001905 | https://www.r-project.org/ |
| Software, algorithm | Fiji | *Schindelin et al., 2012* | RRID:SCR_008606 | https://fiji.sc/ |
| Software, algorithm | blender | | RRID:SCR_008606 | https://www.blender.org/ |
| Software, algorithm | TrakEM2 | | RRID:SCR_008954 | https://imagej.net/TrakEM2 |
| Software, algorithm | CATMAID | | RRID:SCR_006278 | http://www.catmaid.org |
| Software, algorithm | neuroboom | | | https://pypi.org/project/neuroboom/ |

## Fly line and fly rearing

Flies of the genotype *Orco-GAL4; UAS-GCaMP6s* (*Vosshall et al., 2000*) were obtained from the Bloomington *Drosophila* Stock Center (https://bdsc.indiana.edu/) and reared on standard *Drosophila* food at 25°C and 70% humidity on a 12:12 hr day: night cycle. Seven-day-old female flies were used. In these flies, Orco-positive olfactory sensory cells emit green fluorescence, making it possible to identify individual glomeruli.

## Brain dissection and fixation for FIB-SEM

Two 7-day-old female flies were anesthetized with nitric oxide (with Sleeper TAS; INJECT+MATIC, Switzerland) and decapitated with forceps. Heads were dipped for 1 min in 0.05% Triton X-100 in 0.1 M Sorensen's phosphate buffer, pH 7.3 and transferred to a droplet of freshly prepared ice-cooled fixative 2.5% glutaraldehyde and 2.0% paraformaldehyde in 0.1 M Sørensen's phosphate buffer, pH 7.3; as in *Karnovsky, 1965*. The proboscis was removed and the back of the head was opened to improve fixative penetration. After 5–10 min, the brain was dissected out of the head capsule and

post-fixed for 2 hr on ice. Fixation was stopped by rinsing the brain several times in ice-cooled 0.1 M Sørensen's phosphate buffer, pH 7.3 (after *Rybak et al., 2016b*).

## Laser branding of glomeruli for identification during FIB-SEM

To identify the glomeruli of interest at the ultrastructural level and to limit to a minimum the volume of tissue to be scanned with FIB-SEM, near-infrared laser branding NIRB (*Bishop et al., 2011*). Glomeruli of interest were first located with light microscopy in brains of *Orco-GAL4; UAS-GCaMP6s* flies using a confocal microscope (ZEISS LSM 710 NLO, Carl Zeiss, Jena, Germany), a 40x water immersion objective (W Plan-Apochromat 40x/1.0 DIC VIS-IR, Carl Zeiss, Jena, Germany), a laser wavelength of 925 nm at 30% laser power and ZEN software (Carl Zeiss, Jena, Germany). Once glomeruli DA2 or DL5 were identified by means of location, shape, and size, the VOI was tagged with fiducial marks (laser-branded) close to the borders of the glomerulus (*Figure 1A, B*), using an infrared Chameleon Ultra diode-pumped laser (Coherent, Santa Clara, USA) at wavelength 800 nm and at 75–90% of laser power. Two laser scan rounds were performed for each induced fiducial brand. DA2 (right AL) and DL5 (left AL) were laser-branded in the same fly. A second glomerulus DA2 was marked in the right AL of another fly.

## Transmission electron microscopy

Brains were fixed using the Karnovsky's fixative as described above and rinsed with 0.1 M sodium-cacodylate buffer and post-fixed in 1% osmium tetroxide and 1% potassium ferrocyanide in caco-dylate buffer for 2 hr. After rinsing with cacodylate buffer, the brains were dehydrated with a graded acetone series (30–100% acetone), including an additional *en bloc* staining step in-between, in which the brains were incubated in 1% uranyl acetate in 50% acetone for 30 min in the dark and gradually infiltrated with Araldite (glycerol-based aromatic epoxy resins; Serva, Germany). In the final step, the tissue was embedded in pure resin and left in a 60°C incubator to polymerize for 48 hr. Resin blocks were trimmed with a Reichert UltraTrim microtome (Leica, Deer Park, USA), and the fiducial laser marks were then located in semi-thin sections. To check tissue quality before performing high-resolution volume-based EM, serial sections 50 nm in thickness were cut with a diamond knife (Ultra 45°, Diatome, Switzerland) on a Reichert Ultracut S ultramicrotome (Leica, Deer Park, Germany), collected on single slot grids (2 × 1 mm), and imaged with a JEM 1400 electron microscope (Jeol, Freising, Germany) operated at 80 kV. Digital micrographs were obtained with a Gatan Orius SC 1000 CCD camera (Gatan Orius SC 1000; Gatan, Pleasanton, USA) controlled with the Gatan Microscopy Suite software Vers. 2.31.734.0.

## Focused ion beam-scanning electron microscopy

Before serial FIB milling and SEM imaging (*Knott et al., 2008*; *Xu et al., 2017*), the surface of the trimmed block was coated with a conductive carbon layer to prevent charging artifacts. A FEI Helios NanoLab G3 UC (FEI, Hillsboro, USA) was used for FIB-SEM process. The laser marks used to land-mark the VOI were visible across the surface of the block. The VOI surface was protected via a local deposition of platinum using a gas injection system for subsequent ion and electron beam deposition. The material surrounding the VOI at the front and the side was removed to reduce re-deposition of material during FIB-SEM. Serial images across the entire VOI were generated by repeated cycles of milling slices orthogonal to the block surface via FIB and imaging via SEM the newly exposed surface. The tissue was milled with a focused beam of gallium ions using FEI's Tomahawk ion column (accel-erating voltage: 30 kV, beam current: 790 pA, milling steps: 20 nm). FEI's Elstar electron column was used to create the backscattered electron contrast images using an In-Column Detector (accelerating voltage: 3 kV; 1.6 nA; dwell time: 10 µs). The DA2 and DL5 volumes in the first fly were imaged with a resolution of 4.9 × 4.9 × 20 $nm^3$/vox (DA2: 769 images with 4096 × 3536 pix; DL5: 976 images with 5218 × 3303 pix). The volume of a second DA2 in a second fly was imaged with a resolution of 4.4 × 4.4 × 20 $nm^3$/vox (571 images with 4096 × 3536 pix). The milling/imaging cycles were controlled with the Auto Slice and View 4.0 software (FEI, Hillsboro, USA).

## Image alignment, 3D reconstruction, and segmentation

FIB-SEM image stacks were aligned by maximizing the Pearson correlation coefficient of the central part of two consecutive images using template matching from the openCV library (https://imagej.net/

TrakEM2). Dense reconstructions of the glomeruli were produced by manually tracing all neuronal fibers and by annotating all synapses within the two glomeruli, using a skeleton-based reconstruction procedure similar to previous approaches (*Berck et al., 2016*; *Schneider-Mizell et al., 2016*; *Zheng et al., 2018*). Up to five independent tracers and two reviewers participated in an iterative reconstruction process using the web-based reconstruction software CATMAID (http://www.catmaid.org; RRID:SCR_006278; *Saalfeld et al., 2009*; *Schneider-Mizell et al., 2016*; *Figure 1D*, *Figure 1—video 1*), performing a dense reconstruction of synaptic neuropil. In a second fly, neurons of a DA2 glomerulus were manually reconstructed with the volume-based reconstruction method TrakEM2 (*Cardona et al., 2012*), an ImageJ (Fiji) plugin (https://imagej.net/TrakEM2).

## Neuron visualization

Reconstructed neurons were visualized using CATMAID 3D visualization (http://www.catmaid.org) and using Blender 3D, an open-source 3D software (https://www.blender.org/; *Figure 7—figure supplement 1*). Neuron data from CATMAID were imported and shaded by Strahler order using an existing CATMAID plugin for Blender (https://github.com/schlegelp/CATMAID-to-Blender; *Schlegel et al., 2016*). Volume-based reconstructions were visualized as surface shapes in CATMAID imported from TrakEM2 (https://imagej.net/TrakEM).

## Glomerular border definition

The definition of the boundary between olfactory glomeruli was based on the combination of several structural features: the spatial position of pre- and postsynaptic elements along OSN axons, the position of the majority of uPN postsynaptic sites, the faint glial leaflets scattered at the periphery of the glomerulus, and the fiducial laser marks (*Figure 1B, D*).

## Neuron identification

Neuronal fibers were assigned to one of three pre-defined neuron classes: OSNs, uPNs, and MGNs. The classification was based on their 3D shape (*Figure 2A*), their branching intensity (*Figure 2B*), the average diameter of their fibers (neuronal profiles: *Figure 2A* – FIB-SEM image; exemplary volume-based reconstruction), the ratio of T-bars-to-input sites and the size of their T-bars, which were either 'small' (few postsynaptic connections) or 'large' (many postsynaptic connections *Figure 2—figure supplement 1*). In addition, several intracellular features helped to classify neuron classes: the shape and appearance of mitochondria, the size and electron density of vesicles, and the amount of synaptic spinules (small filopodia-like invaginations of neighboring cells) (*Figure 2A* – FIB-SEM image; *Gruber et al., 2018*). OSNs and uPNs could be counted, due to their uniglomerular character, by means of the identification of the axons (OSNs) or main dendrites (uPNs) entering the glomerulus. The number of MGNs could not be counted because of their pan-glomerular projection patterns in the AL. Ipsi- and contralateral OSNs in DA2 and DL5 were identified based on the trajectory of axonal fibers and their entry location in each glomerulus (example neurons: *Figure 4B*). Ipsilateral OSNs reach the glomerulus from the ipsilateral antennal nerve and leave the glomerulus toward the antennal lobe commissure (ALC: *Tanaka et al., 2012*). Contralateral OSNs reach the glomerulus projecting from the ALC.

## Data analysis

With the aid of the web-based software CATMAID (http://www.catmaid.org/) the following properties were quantified: glomerular volume, neuronal fiber length (in µm), number of fiber branching points, number of synaptic input and output sites, and T-bars (see data availability). In a second fly, the volume of neurons in DA2 was measured (*Figure 2—figure supplement 1*) with the aid of TrakEM2 (*Cardona et al., 2012*), an ImageJ (Fiji) plugin (https://imagej.net/TrakEM2). The following calculations were performed:

1. Innervation density $= \frac{\text{total neuron length } (\mu m)}{\text{glomerular volume } (\mu m^3)}$

   a. Calculated as a ratio: (1) the sum of all neuronal fibers of each neuron class or (2) all together (*Table 1*) or (3) for each neuron individually (*Figure 3*).

2. *Glomerular synaptic density* $= \frac{\text{\# of synaptic inputs}, - \text{ outputs or } eT-bars}{\text{glomerular volume } (\mu m^3)}$

 a. Calculated as a ratio: (1) the sum of all neuronal fibers of each neuron class or (2) all together (*Table 1*) or (3) for each neuron individually (*Figure 3*)

3. $Neuronal\ synaptic\ density = \frac{\#\ of\ synaptic\ inputs-,outputs\ or\ T-bars}{neuronal\ fiber\ length\,(\mu m)}$ (*Table 1*; *Figure 3—figure supplement 1*)

4. $Synaptic\ ratios = \frac{\#\ of\ T-bars\ or\ outputs}{inputs}$ (represents the average for each neuron class; *Table 1*)

5. $Polyadicity = \frac{\#\ of\ outputs}{T-bars}$ (represents the average number of postsynaptic sites at a T-bar of each neuron class; *Table 1* and *Figure 1E*)

6. $Relative\ differences = \frac{respective\ value\ target\ glomerulus-value\ source\ glomerulus}{source\ glomerulus} \times 100$ (*Supplementary file 1*; *Figure 5—source data 1*)

7. Relative synaptic strength $= \frac{\#\ of\ synaptic\ contacts\ neuron\ class\ A\ to\ B}{\#all\ synaptic\ contacts\ in\ corresponding\ glomerulus}$ (*Supplementary file 1*; *Figure 5—source data 1*)

8. Fraction of output $= \frac{\#\ of\ outputs\ of\ neuron\ class\ A\ directed\ to\ neuron\ class\ B}{total\ \#\ of\ outputs\ of\ neuron\ class\ A} \times 100$

9. Fraction of input $= \frac{\#\ of\ inputs\ neuron\ class\ A\ from\ class\ B}{total\ \#\ of\ inputs\ of\ neuron\ class\ A} \times 100$

Graphs were made with the programming language R and RStudio (*R Development Core Team, 2018*) using the packages 'ggplot2' and 'reshape' (https://www.r-project.org/) or with Python (see data availability). EM and fluorescence images were visualized with ImageJ (Fiji) (http://fiji.sc/; *Schindelin et al., 2012*) and All figures were compiled with Adobe Illustrator CS5 software (Adobe Inc).

Statistical analysis was performed with R Studio (*RStudio Team, 2016*) using the packages 'ggsignif' (https://www.r-project.org/). Differences between samples DA2 and DL5 or between ipsilateral and contralateral OSNs were tested for significance with a two-sided Student's *t*-test if sample size was normally distributed, or with Wilcoxon two-sample test if the data was not normally distributed (noted in figure legend). Data is in all cases represented as mean + standard deviation.

## Analysis of autapses

The location of autapses, the measurement of their geodesic distance (distance along the neuronal dendrite), and the number of branching points from point A (presynaptic site) to B (postsynaptic profile) were analyzed with Python using the package 'neuroboom' (*Pleijzier, 2022*; see also data availability).

## Acknowledgements

The authors are most grateful to Katrin Buder (Leibniz Institute on Aging, Jena) for the support with electron microscopy sample preparation, and Veit Grabe (MPI for Chemical Ecology, Jena) for advice on two-photon imaging. Great thanks also to Albert Cardona for discussion on synaptic networks, him, and Tom Kazimiers (Kazmos GmbH, Dresden) for instruction in the use of CATMAID. The neuronal reconstructions were conducted with the outstanding support of Damilola E Akinyemi, Eckard E Schumann, and Michael Adewoye (MPI for Chemical Ecology, Jena). We thank Martin Nawrot and Magdalena Springer (University of Cologne) for constructive comments and discussions about autapses. The work was supported by Roland Kilper and Ute Müller (aura optik GmbH, Jena), the European Regional Development Fund, by funds from the DFG (grant no. 430592330), in the Priority Program 'Evolutionary Optimization of Neuronal Processing' (DFG-SPP 2205) and by the Max Planck Society.

## Additional information

### Funding

| Funder | Grant reference number | Author |
| --- | --- | --- |
| European Regional Development Fund | aura-optics | Lydia Gruber |
| Deutsche Forschungsgemeinschaft | DFG-SSP 2205 | Jürgen Rybak |

| Funder | Grant reference number | Author |
|---|---|---|
| Deutsche Forschungsgemeinschaft | grant no. 430592330 | Jürgen Rybak |
| Max Planck Society | | Lydia Gruber<br>Martin Niebergall<br>Bill S Hansson<br>Jürgen Rybak |

The funders had no role in study design, data collection, and interpretation, or the decision to submit the work for publication. Open access funding provided by Max Planck Society.

## Author contributions

Lydia Gruber, Conceptualization, Resources, Data curation, Software, Formal analysis, Supervision, Validation, Investigation, Visualization, Methodology, Writing – original draft, Project administration, Writing – review and editing; Rafael Cantera, Conceptualization, Formal analysis, Supervision, Validation, Investigation, Methodology, Writing – review and editing; Markus William Pleijzier, Software, Formal analysis, Validation, Investigation, Visualization, Methodology, Writing – review and editing; Martin Niebergall, Data curation, Software, Validation, Investigation; Michael Steinert, Resources, Data curation, Software, Formal analysis, Investigation, Visualization, Methodology, Writing – review and editing; Thomas Pertsch, Resources, Software, Investigation, Visualization, Methodology, Writing – review and editing; Bill S Hansson, Conceptualization, Resources, Software, Supervision, Funding acquisition, Project administration, Writing – review and editing; Jürgen Rybak, Conceptualization, Resources, Data curation, Software, Formal analysis, Supervision, Funding acquisition, Validation, Investigation, Visualization, Methodology, Writing – original draft, Project administration, Writing – review and editing

## Author ORCIDs

Lydia Gruber ![ORCID] https://orcid.org/0000-0002-3882-2603
Rafael Cantera ![ORCID] https://orcid.org/0000-0002-4990-3898
Markus William Pleijzier ![ORCID] https://orcid.org/0000-0002-7297-4547
Martin Niebergall ![ORCID] https://orcid.org/0000-0002-2869-6949
Michael Steinert ![ORCID] https://orcid.org/0000-0001-9197-6305
Thomas Pertsch ![ORCID] https://orcid.org/0000-0003-4889-0869
Bill S Hansson ![ORCID] https://orcid.org/0000-0002-4811-1223
Jürgen Rybak ![ORCID] https://orcid.org/0000-0003-0571-9957

Reviewer #1 (Public review): https://doi.org/10.7554/eLife.88824.3.sa1
Reviewer #2 (Public review): https://doi.org/10.7554/eLife.88824.3.sa2
Author response https://doi.org/10.7554/eLife.88824.3.sa3

# Additional files

## Supplementary files

Supplementary file 1. Relative differences of innervation and synaptic composition between glomeruli DA2 and DL5. The table lists the relative differences between DA2 and DL5 (see Methods for calculations). Values that are at least 20% greater in DA2 than in DL5 are highlighted in dark shades and values that are at least 20% less in DA2 than in DL5 are highlighted in light shades.

MDAR checklist

## Data availability

Full datasets for DA2 and DL5 (X_DA2 -rAL, X_DL5-lAL) and the partial dataset of DA2 (VII_DA2-rAL) are available through https://doi.org/10.17617/3.WL9UQN. These datasets include the FIB-SEM image set (as tar files), the assembly of all neurons, connectors, tafs, and annaotations (in JSON format), and a header file (in yaml format). The data for all glomeruli of this study can also be assessed on a public CATMAID instance: https://catmaid.ice.mpg.de/catmaid_2020.02.15. Neurons are named according to their neuron classification. The neuroboom Python package was used for dendrogram

analysis, available at https://github.com/markuspleijzier/neuroboom (*Pleijzier, 2022*) and https://pypi.org/project/neuroboom/. Further source code used for data analysis is accessible via GitHub (https://github.com/markuspleijzier/Gruber_eLife; copy archived at *Pleijzier, 2025*).

The following dataset was generated:

| Author(s) | Year | Dataset title | Dataset URL | Database and Identifier |
|---|---|---|---|---|
| Lydia G, Jürgen R | 2025 | Focused-Ion-Beam Scanning Electron Microscopy (FIB-SEM) based image dataset of antennal lobe glomeruli in *Drosophila melanogaster* | https://doi.org/10.17617/3.WL9UQN | Edmond, 10.17617/3.WL9UQN |

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
