## [Editor Report · eLife Assessment]

This study seeks to determine how synaptic relationships between principal cell types in the olfactory system vary with glomerulus selectivity and is therefore **valuable** to the field. The methodology is **solid**, and with the caveat that here was a technical need to group all local interneurons, centrifugal neurons and multiglomerular projection neurons into one category ("multiglomerular neurons"), this work reveals some very interesting potential differences in circuit architecture associated with glomerular tuning breadth.

---

## [Referee Report · Reviewer #1 (Public review)]

In this manuscript, Gruber et al perform serial EM sections of the antennal lobe and reconstruct the neurites innervating two types of glomeruli - one that is narrowly tuned to geosmin and one that is broadly tuned to other odours. They quantify and describe various aspects of the innervations of olfactory sensory neurons (OSNs), uniglomerlular projection neurons (uPNs), and the multiglomerular Local interneurons (LNs) and PNs (mPNs). They find that narrowly tuned glomeruli had stronger connectivity from OSNs to PNs and LNs, and considerably more connections between sister OSNs and sister PNs than the broadly tuned glomeruli. They also had less connectivity with the contralateral glomerluli. These observations are suggestive of strong feed-forward information flow with minimal presynaptic inhibition in narrowly tuned gomeruli, which might be ecologically relevant, for example, while making quick decisions such as avoiding a geosmin-laden landing site. In contrast, information flow in more broadly tuned glomeruli show much more lateralisation of connectivity to the contralateral glomerulus, as well as to other ipsilateral glomeruli.

The data are well presented, the manuscript clearly written, and the results will be useful to the olfaction community. I had earlier suggested comparisons with other EM datasets that exist to investigate stereotypy, and am convinced by their efforts and reasons for which these were either not possible to do or not possible within the timeframe of a revision.

Comments on revisions:

Thank you for the careful responses to my suggestions. I hope that such approaches will be possible by others going forward.

---

## [Referee Report · Reviewer #2 (Public review)]

The chemoreceptor proteins expressed by olfactory sensory neuron differ in their selectivity such that glomeruli vary in the breadth of volatile chemicals to which they respond. Prior work assessing the relationship between tuning breadth and the demographics of principal neuron types that innervate a glomerulus demonstrated that narrowly tuned glomeruli are innervated more projection neurons (output neurons) and fewer local interneurons relative to more broadly tuned glomeruli. The present study used high resolution electron microscopy to determine which synaptic relationships between principal cell types also vary with glomerulus tuning breadth using a narrowly tuned glomerulus (DA2) and a broadly tuned glomerulus (DL5). The strength of this study lies in the comprehensive, synapse-level resolution of the approach. Furthermore, the authors implement a very elegant approach of using a 2-photon microscope to score the upper and lower bounds of each glomerulus thus defining the bounds of their restricted regions of interest. Using the approach, the authors identify several architectural motifs that differ between glomeruli with different tuning properties

In the revised version of this study the authors discuss several important limitations. There was a technical need to group all local interneurons, centrifugal neurons and multiglomerular projection neurons into one category ("multiglomerular neurons") which complicates interpretations as even multiglomerular projection neurons are very diverse. With only 2 narrowly tuned glomeruli and 1 broadly tuned glomerulus, architecture differences may reflect more than just differences in tuning breadth. Finally, the degree to which inter-animal variability may contribute to differences between glomeruli is discussed. If these caveats are kept in mind, this work reveals some very interesting potential differences in circuit architecture associated with glomerular tuning breadth.

This work establishes specific hypotheses about network function within the olfactory system that can be pursued using targeted physiological approaches. It also identifies key traits that can be explored using other high resolution EM datasets and other glomeruli that vary in their tuning selectivity. Finally, the laser "branding" technique used in this study establishes a reduced cost procedure for obtaining smaller EM datasets from targeted volumes of interest by leveraging the ability to transgenically label brain regions in Drosophila.

Comments on revisions:

I appreciate the thoughtful responses that the authors made regarding the initial assessment of their study. The authors discuss these limitations in their manuscript which should not be viewed as criticisms, but rather caveats to be considered for this study specifically and in some instances, for all connectomics studies.

I still believe there is a lost opportunity to make use of the FlyWire dataset to make specific strategic comparisons. I do not propose attempting to replicate the comprehensive nature of the main study, but querying cell type based on glomerular innervation would allow the authors to address consistency of observed differences between glomeruli as ORNs and uPNs have been thoroughly annotated and analysis can be limited by neuropil. I agree that it is unclear how many individuals would need to be examined to achieve sufficient statistical power, but some of the circuit motifs revealed in this study can be readily tested in the FlyWire dataset. For instance, the observation from this study that narrowly tuned ORNs receive less synaptic input from LNs is supported in FlyWire, with DL5 ORNs getting far more synaptic input from LNs relative to DA2 and VA1v. I'm not proposing repeating all of the analyses from this study, and there is no doubt that inter-animal variability and technical differences can explain different observations across datasets, but I believe these are considerations of which the readers (who can query these synaptic relationships in FlyWire) should be made aware.

---

## [Author Response]

The following is the authors’ response to the original reviews.

**Reviewer #1 (Public Review):**
In this manuscript, Gruber et al perform serial EM sections of the antennal lobe and reconstruct the neurites innervating two types of glomeruli one that is narrowly tuned to geosmin and one that is broadly tuned to other odours. They quantify and describe various aspects of the innervations of olfactory sensory neurons (OSNs), uniglomerlular projection neurons (uPNs), and the multiglomerular Local interneurons (LNs) and PNs (mPNs). They find that narrowly tuned glomeruli had stronger connectivity from OSNs to PNs and LNs, and considerably more connections between sister OSNs and sister PNs than the broadly tuned glomeruli. They also had less connectivity with the contralateral glomeruli. These observations are suggestive of strong feed-forward information flow with minimal presynaptic inhibition in narrowly tuned glomeruli, which might be ecologically relevant, for example, while making quick decisions such as avoiding a geosmin-laden landing site. In contrast, information flow in more broadly tuned glomeruli show much more lateralisation of connectivity to the contralateral glomerulus, as well as to other ipsilateral glomeruli.The data are well presented, the manuscript clearly written, and the results will be useful to the olfaction community. I wonder, given the hemibrain and FAFB datasets exist, whether the authors have considered verifying whether the trends they observe in connectivity hold across three brains? Is it stereotypic?

We appreciate the reviewer’s positive view of our study and their thoughtful and relevant comment on the issue of individual variation. We agree in that this is a very important question and notice that it was also asked for by the second Reviewer. It reflects both our limited understanding of the range of individual variation in synaptic connectivity—whether in flies, humans, or other species—and the challenge of determining which of the differences observed in our study are stereotypical features of each glomerulus type. Undoubtedly this criticism addresses a crucial problem of practically all connectome studies so far and for which there is no immediate solution. This type of studies requires so much time, efforts and money that increasing the number of samples is seldom feasible. The Reviewer wonders if we could compare our data with that made available by two of the largest connectome studies of Drosophila. This appeared to us to be a very good idea and we have tried to follow the advice but, unfortunately, it was impracticable because of the reasons we explain below. The hemibrain data cannot be used for this purpose because it does not contain the full glomerulus DA2 (Schlegel et al., 2021). A different problem hindered us from using the FAFB dataset, the other dataset mentioned by the Reviewer. In this case the three glomeruli were sectioned and reconstructed but the dataset lacks an annotated list of all synaptic connections corresponding to each glomerulus. Such annotation (a compendium of all synaptic connections inside each glomerulus informing for each connection which type of neuron provides the presynaptic site and which the postsynaptic site) is essential for direct comparison with our data. It is important to keep in mind that the current analytical tools available for the use of these datasets (e.g., NeuPrint, FlyWire and CATMAID) do not offer the ability to extract data on synapses exclusively from the glomerular volume of DA2 or DL5. In this case, it certainly is theoretically possible to obtain the data by doing ourselves the annotation. However, such a study will demand so much time, efforts and financial resources, which we believe would not be justified solely to increase the number of individuals from one to two. Instead, our manuscript includes a comparison of the OSN connectivity in VA1v and DL5 using the hemibrain dataset published by Schlegel et al. (2021) (see revised manuscript: lines 311–315; 431–434; 558–562; 602–606).

Beyond the opinion, that we share in full with the Reviewer, that a comparison including three flies will be better than a comparison made with one glomerulus of each type we are still challenged by the question of which -if any- of the differences are stereotypic. The clarification of what are stereotypical differences between particular glomeruli in features as those discussed in our study and what is simply differences within the normal range of individual variation is basically a statistical problem. A first attempt at a comprehensive comparison focusing on intra- and inter-individual variability was recently made by comparing two connectome datasets from two different Drosophila individuals (Dorkenwald et al., 2024; Schlegel et al., 2024). At present, it is still unclear how many samples are needed to make a statistically robust comparison of olfactory synaptic circuits in adult flies—perhaps 3, 6, or even 18 individuals?

**Reviewer #2 (Public Review):**
The chemoreceptor proteins expressed by olfactory sensory neurons differ in their selectivity such that glomeruli vary in the breadth of volatile chemicals to which they respond. Prior work assessing the relationship between tuning breadth and the demographics of principal neuron types that innervate a glomerulus demonstrated that narrowly tuned glomeruli are innervated more projection neurons (output neurons) and fewer local interneurons relative to more broadly tuned glomeruli. The present study used high-resolution electron microscopy to determine which synaptic relationships between principal cell types also vary with glomerulus tuning breadth using a narrowly tuned glomerulus (DA2) and a broadly tuned glomerulus (DL5). The strength of this study lies in the comprehensive, synapse-level resolution of the approach. Furthermore, the authors implement a very elegant approach of using a 2-photon microscope to score the upper and lower bounds of each glomerulus, thus defining the bounds of their restricted regions of interest. There were several interesting differences including greater axo-axonic afferent synapses and dendrodentric output neuron synapses in the narrowly tuned glomerulus, and greater synapses upon sensory afferents from multiglomerular neurons and output neuron autapses in the broadly tuned glomerulus. The study is limited by a few factors. There was a technical need to group all local interneurons, centrifugal neurons, and multiglomerular projection neurons into one category ("multiglomerular neurons") which complicates any interpretations as even multiglomerular projection neurons are very diverse. Additionally, there were as many differences between the two narrowly tuned glomeruli as there were comparing the narrowly and broadly tuned glomeruli. Architecture differences may therefore not reflect differences in tuning breadth, but rather the ecological significance of the odors detected by cognate sensory afferents. Finally, some synaptic relationships are described as differing and others as being the same between glomeruli, but with only one sample from each glomerulus, it is difficult to determine when measures differ when there is no measure of inter-animal variability. If these caveats are kept in mind, this work reveals some very interesting potential differences in circuit architecture associated with glomerular tuning breadth.This work establishes specific hypotheses about network function within the olfactory system that can be pursued using targeted physiological approaches. It also identifies key traits that can be explored using other high-resolution EM datasets and other glomeruli that vary in their tuning selectivity. Finally, the laser "branding" technique used in this study establishes a reduced-cost procedure for obtaining smaller EM datasets from targeted volumes of interest by leveraging the ability to transgenically label brain regions in Drosophila.

CLASSIFICATION OF NEURONAL TYPES

We agree that grouping diverse types of interneurons into a single category (referred to as MGNs) limits the ability to make interpretations about synaptic similarities and differences between specific neuronal types. This was, however, an unavoidable compromise resulting from our decision to generate a comprehensive, synapse-level reconstruction of the restricted regions encompassing the DA2 and DL5 glomeruli. As both reviewers have noted, this approach offers significant value and we hope the Editor will also recognize that this limitation does not prevent readers from gaining important and novel insights into the synaptic circuitry of these two glomeruli.

Similar to the approach taken by Tobin at al. (2017) we prioritized producing a densely reconstructed neuropile, in which no synapses were omitted (Tobin et al., 2017). The downside of this method is that not all synaptic connections could be reliably assigned to specific neuronal types, with about 12% remaining unassigned." We anticipate that future research, supported by advances in semi-automated tracing methods, improved imaging technologies, and increased personnel resources, will allow not only for the generation of more complete connectomes of the entire brain (Scheffer et al., 2020; Zheng et al., 2018), but also, for the accurate reconstruction and classification of individual synapses—even in highly complex regions such as the olfactory glomeruli. We also expect that a second complete connectome of a male Drosophila will soon become available, which will provide valuable opportunities for comparisons across individuals and between male and female brains in future studies.

INTERGLOMERULAR DIFFERENCES

Thank you for this insightful comment. It is indeed true that despite both DA2 and VA1v being narrowly tuned glomeruli, they exhibit considerable differences in specific connectivity features (e.g., relative synaptic strengths above certain thresholds) and that those differences can be as pronounced as those observed between DA2 and the broadly tuned DL5. For this reason, comparing each individual glomerulus to every other is not a practical or informative approach. To derive robust interpretations, we focused instead on whether two glomeruli that share a particular functional characteristic—namely, being narrowly tuned for single odorants—also share connectivity patterns that distinguish them from a broadly tuned reference glomerulus.

Our results support this. Furthermore, additional connectomics data reinforce our conclusions.

For example, OSN-OSN connectivity is stronger in the two narrowly tuned glomeruli (DA2 and VA1v) relative to the broadly tuned glomerulus (DL5). While these pairwise differences alone are not conclusive, the finding that the two narrowly tuned glomeruli studied here share features that distinguish them from the broadly tuned glomerulus supports our interpretation. We found further support for this idea in the data reported by Schlegel et al. (2021) further. In that dataset, other narrowly tuned glomeruli (DA1, DL3, and DL4) also exhibit stronger OSNOSN connectivity than other broadly tuned glomeruli (DM1 or DM4).

We do not deny that there are many differences between any given pair of glomeruli, regardless of whether they are narrowly or broadly tunned. Instead, we propose that our findings on circuit features indicate that most of the observed differences actually grouped the two narrowly tuned glomeruli together relative to the broadly tuned glomerulus. A more concise summary is now provided in the newly added Figure 8. We also added explanatory lines of text in the beginning of the chapter ‘specific features of narrowly tuned glomerular circuits.

ECOLOGICAL SIGNIFICANCE

This is an interesting point. However, it is difficult to disentangle the "ecological significance" of processed odorants from the "tuning breadth" of a glomerulus. In the Drosophila olfactory system, glomerular circuits that respond to ecologically important odorants—such as those involved in reproduction or danger—tend to be more narrowly tuned. Moreover, while we refer to odorants with specific ecological significance as those linked to survival or reproductive behaviors, defining the significance of an odorant with precision is inherently challenging, as it can vary depending on context and environmental conditions.

What both circuits share is their narrow tuning breadth. We therefore propose that the common circuit features of VA1v and DA2, highlighted in this study, are functionally related to the fact that each circuit processes single odorants. Consequently, their specificity is most likely determined at the level of the receptor.

INDIVIDUAL VARIABILITY

We agree that accounting for inter-animal variability would strengthen the study. However, we are confident that even a modest statistically sound assessment of this variability would require a larger sample size, certainly more than just two or three flies, which is presently not feasible.

We refer the reviewer to our response to Reviewer #1 regarding this important issue.

Initial insights into variability between flies have been provided through comparative analyses of the two most comprehensive female *Drosophila melanogaster* connectomes—the FAFB and hemibrain datasets (Schlegel et al., 2024). For more detailed quantitative comparisons regarding inter-animal variability, please refer to our response to the second major point raised by Reviewer #2. As highlighted by Schlegel et al. (2024), making definitive statements about the stereotypy of neuron numbers, unitary cell-cell connections (edges), or synaptic strengths (weights) remains a complex challenge."

While appreciating the rigour of this work we were surprised to notice the omission of a comparison of their observations with the two other existing datasets. This would not only have addressed the technical limitation of this particular study - the inability to identify specific neuron types due to imaging a small part of the brain - but would also have shed light on inter-animal variability

We strongly recommend that the authors do make this comparison - the datasets are currently extremely user friendly and so we don't estimate the replication of their key findings will be too onerous. This will be particularly important to resolve the issue of having to classify all multiglomerular local interneurons and multiglomerular projection neurons - broadly into "MGN. Such a comparison will dramatically strengthen this study that poses very interesting questions, but in its current form, has this striking shortcoming.

INDIVIDUAL VARIABILITY AS EXPRESSED HERE:

Earlier on we were of the same opinion that the Reviewer express here but, unfortunately, it was not possible to follow his advice. As far as it was possible, we have compared some of our results to the values of the two datasets that the Reviewer refers to, but the absence of glomerulus DA2 in one of the datasets and the absence of synapse annotation for all the relevant glomeruli in the other dataset prevented us from making a full comparison. Moreover, believe that the problem of individual variation most probably cannot be solved by increasing the comparison with one or two more flies.

**Reviewer #1 (Recommendations for The Authors):**
The lines 270 - 282 confused me in the backdrop of Figure 3B.

The concern may stem from our inclusion of a comparison between the uPNs of glomerulus DA2 and the single uPN of glomerulus DL5 in the statistical analysis presented in Figure 3. This comparison was included to ensure a comprehensive representation of the data, highlighting the variability across all major cell groups. We have clarified this rationale in the revised manuscript (see lines 274-282).

**Reviewer #2 (Recommendations for The Authors):**
I commend the authors for taking such a thorough approach to advance an interesting topic in olfaction. The following suggestions are intended to strengthen this study:Major points:A color-blind-friendly palette should be used for all figures. Currently, five of seven figures use red and green, and in particular, Figure 5 will be uninterpretable for red/green color-blind readers.

We are thankful for this important comment. We changed the color palette as suggested by the reviewer, and replaced Red with Magenta and changed the figure legend accordingly.

This level of analysis is extremely resource and time-consuming, so even obtaining this information at this resolution is an impressive achievement. However, this study would be well served by strategically supplementing the analysis of this dataset with information from other publicly available connectomics datasets. For instance, some interpretations are limited because there is information from only a single DL5 and DA2 glomerulus. Any claims in which one glomerulus has more, less, or the same of a metric must be tempered because without replicates, there are no measures of inter-animal variability. As an example, on lines 386-387 the authors state "The relative synaptic strength between MGN>uPN was stronger in DA2 (12%) than DL5 (10%)". It is difficult to assess whether this represents a difference that is outside of the range of inter-animal variability inherent to the olfactory system. Taking select measures from the Hemibrain and FAFB (via FlyWire) datasets could help strengthen these claims.

We fully agree with the Reviewer’s opinion that since our data is from one glomerulus of each type “It is difficult to assess whether this represents a difference that is outside of the range of inter-animal variability inherent to the olfactory system.” This is a weakness of practically all connectome studies based on electron microscopy in both Drosophila and other animals We cannot be sure that measurements from the Hemibrain and FAFB datasets could help strengthen our claims, because the magnitude of the range of individual variation is presently not known and most probably solving this problem will require more than one or two more flies. In any case, it is not possible to follow this advice and compare our data with that of the hemibrain because the DA2 was not included in that study. We ask the Reviewer to read our more detailed explanation in our response to Reviewer 1.

In the particular case commented by the Reviewer above, the relative difference in synaptic strength exceeds 20%. Whether such a difference has functional relevance remains an open question but Schlegel et al. (2024) support our interpretation. They showed that synaptic weights with differences larger than 20% tend to be consistent across individuals, with strong correlations within and between animals (Pearson’s R = 0.97 and R = 0.8; Fig. 4).

Grouping all local interneurons, centrifugal neurons response and multiglomerular PNs into one category limits the ability to make interpretations about similarities or differences in the synaptic relationships involving MGNs. The authors could get an estimate of the number of multiglomerular PNs in DL5, VA1v, and DA2 from Hemibrain and FlyWire platforms to get a better sense of differences between glomeruli in the MGN category.

We agree in that grouping a variety of interneurons into a single category (called MGNs) limits the ability to make interpretations about similarities or differences in the synaptic relationships involving different neurons. This was the unavoidable price to be paid once we decided to register a “comprehensive, synapse-level resolution” map of these two glomeruli. It appears to us that both reviewers have clearly recognized the intrinsic value of this approach and we hope that the Editor will share this opinion.

Consistent with the assumptions of Tobin et al., (2017) our hypothesis on LN connectivity differences is based on the fact that they are the most numerous and broadly arborizing neurons of the class that we call multiglomerular neurons in the AL (Chou et al., 2010; Lin et al., 2012; Tanaka et al., 2012). Recent connectome studies confirm this feature across all glomeruli (Bates et al., 2020; Horne et al., 2018; Scheffer et al., 2020; Schlegel et al., 2021; Zheng et al., 2018).

In response to the reviewer’s question, we conducted a case-specific reanalysis of the data from Horne (2018), which provides comprehensive connectivity information for the VA1v glomerulus. This allowed us to quantify the proportional contributions of LNs (n = 56) and mPNs (n = 13) to all MGN connections (MGN-MGN, MGN>OSN, MGN>uPN, uPN>MGN, OSN>MGN).

Our analysis showed that 84% of MGN output originates from LNs. 57% of the input to MGN comes from LNs and 43% from mPNs, largely due to strong OSN>mPN input. Thus, for the filtered MGN connections relevant to distinguishing narrowly from broadly tuned circuits (e.g., MGN>OSN, uPN>MGN; see Fig. 8), LNs are the dominant contributors in VA1v. (These data are not included in the resubmitted manuscript.) This supports our interpretation that the LN are responsible for the majority of MGN connections underlying the observed differences between glomeruli.

For instance, prior work has reported fewer local interneurons innervating DA2, but in this study there was an unexpected result that there was greater MGN innervation density and synapse # for DA2 relative to DL5 This discrepancy could be due to differences in the number of multiglomerular PNs innervating each glomerulus, which would be obscured when these PNs are combined with local interneurons in the MGN category.

"We agree that the greater MGN innervation density in DA2 in our study could reflect a stronger contribution from mPNs. However, innervation density alone does not indicate how many mPNs actually innervate DA2 or DL5. Alternatively, increased innervation and/or synaptic frequency of local interneurons (LNs) could also account for this observation. In our view, neuron number does not necessarily correlate with branching complexity or synaptic density.

For example, the dendritic length of the single uPN in glomerulus DL5 is approximately equal to the combined dendritic length of the multiple uPNs of the DA2. Similarly, Tobin et al. (2017) reported that when comparing uPNs in glomerulus DM6 between the left and right brain hemispheres, they found variability in cell number but not in dendritic length. More recently, the FAFB and hemibrain datasets showed a similar pattern in another neuronal type. A substantial variation in cell number was observed for Kenyon cells between the two Drosophila individuals, but this cell type consistently makes and receives, in both individuals, similar presynapses and post-synapses (Schlegel et al., 2024).

On line 33 the authors cannot claim that DA2-OSNs experience less presynaptic inhibition based on the data in this study. Even without the limitations of the MGN category (described above), presynaptic inhibition depends on more than just the number of synapses, rather it is affected by GABA B receptor expression levels and the second messenger components downstream of this receptor. Physiological experiments are needed to justify this claim, so I recommend adjusting accordingly.

We agree with the Reviewer and have adjusted the text on line 33 and in the main body of the text by referring to this finding as “presynaptic input”, which is what we have quantified, instead of “less presynaptic inhibition”.

Figures 5 and 6 seek to distill the wealth of information from this study into broad takehome points for the reader, while still providing a good amount of detail. I think a final more concise graphic summary (similar to the graphical abstract or Figure 6 of Grabe et al 2016) depicting the most critical differences between glomeruli would further clarify the broad findings of this study.

We appreciate this comment and we have added a “graphic summary” as the Reviewer proposed. We made a new figure that becomes Figure 8 and summarizes our results and highlights differences between narrowly and broadly tuned glomeruli in a more concise graphical abstract format.

Minor points:Much of the manuscript provides details about synapse fractions or % synapses for a given synaptic relationship. Please ensure that it is clear which principal cell types are being described, as it can be easy to get lost. - Should line 284 say "...than DL5 as it has been reported that DA2 is innervated by fewer LNs..."?

We appreciate the reviewer’s comment and we have corrected this sentence that now reads as follows: (see text: beginning at line 290).

Taisz et al. has been published, so the citation should be updated.

We have updated the corresponding citation.

On line 233, the authors ascribe the small electron-dense vesicles as likely housing sNPF released by MGNs. However, Carlsson et al. (2010) demonstrated that sNPF is released by OSNs, which was further functionally characterized by Root et al. (2011) and Ko et al. (2014). In terms of MGNs that release neuropeptides, Carlsson et al. 2010 demonstrated that local interneurons immunolabel for tachykinin, myoinhibitory peptide, and allatostatin-A, while two extrinsic neurons release SIFamide. In theory, aminergic neurons could also have small electron-dense vesicles, but this can be variable.

The Reviewer is completely right in his criticism. The MGN certainly contain neurons that have been reported to contain neuropeptides other than sNPF. We have corrected this sentence and it now reads as follows (page7, line 236): “Interestingly, besides the abundant clear small vesicles..

On line 636, the Berck and Schlegel studies demonstrated that panglomerular local interneurons synapse upon OSN, but not that they induce presynaptic inhibition (which was demonstrated in the studies cited in the next sentence). I recommend adjusting this sentence.

We agree and we have corrected the text following the Reviewers advice. It now reads as follows (page 19. Line 663): “We also observed that OSNs received less MGN feedback.